



# Hydraulic fracture propagation in a heterogeneous stress field in a crystalline rock mass

Nathan Dutler[1], Benoît Valley[1], Valentin Gischig[2], Linus Villiger[3], Hannes Krietsch[3], Joseph Doetsch[3], Bernard Brixel[3], Mohammadreza Jalali[4], and Florian Amann[4]

[1]Center for Hydrogeology and Geothermics (CHYN), University of Neuchâtel, Neuchâtel, Switzerland
[2]CSD Engineers, Bern, Switzerland
[3]Department of Earth Science, ETH Zurich, Zurich, Switzerland
[4]Department of Engineering Geology and Hydrogeology, RWTH Aachen, Aachen, Germany

**Correspondence:** Nathan Dutler (nathan.dutler@unine.ch)

**Abstract.** As part of the In-situ Stimulation and Circulation (ISC) experiment, hydraulic fracturing (HF) tests were conducted in a moderately fractured crystalline rock mass at the Grimsel Test Site (GTS), Switzerland. The aim of these injection tests was to improve our understanding of processes associated with high-pressure fluid injection. A total of six HF experiments were performed in two inclined boreholes, where the surrounding rock mass was accessed with twelve observation boreholes, which

allow high-resolution monitoring of fracture fluid pressure, strain and micro-seismicity in an exceptionally well-characterized rock mass. A similar injection protocol was used for all six experiments to investigate the complexity of the fracture propagation processes. At the borehole scale, these processes involved newly created tensile fractures intersecting the injection interval while at the cross-hole scale, the natural network of fractures dominated the propagation process. The six HF experiments can be divided into two groups based on their injection location (i.e., south or north to a brittle ductile shear zone), their similarity

of injection pressures and their response to deformation and pressure propagation. The injection tests performed in the south connect upon propagation to the brittle ductile shear zone. Thus, the shear zone acts as a dominant drain and a constant pressure boundary. The experiments executed north of the shear zone, show smaller injection pressures and larger backflow during bleed-off phases. From a seismic perspective, the injection tests show high variability in seismic response independent of the location of injection. For two injection experiments, we observe re-orientation of the seismic cloud as the fracture

propagated away from the wellbore. In both cases, the main propagation direction is normal to the minimum principal stress direction. The re-orientation during propagation is interpreted to be related to a strong stress heterogeneity and the intersection of natural fractures striking different than the propagating hydraulic fracture. The seismic activity was limited to about 10 m radial distance from the injection point. In contrast, strain and pressure signals reach further into the rock mass indicating that the process zone around the injection point is larger than the zone illuminated by seismic signals. Furthermore, strain signals

indicate not just single fracture openings but also the propagation of multiple fractures. Transmissivities of injection intervals increase about 2-4 orders of magnitudes.



# 1 Introduction

Hydraulic fracturing (HF) is a technology based on the initiation and propagation of tensile cracks in rock from a wellbore using high-pressure fluid injections. It is used to promote fluid flow through newly-created permeable fractures with the goal to extract energy (heat or hydrocarbons) from the subsurface in formations with insufficient natural permeability (Economides and Nolte, 2000). Massive hydraulic fracturing technology is often used in the oil and gas industry and in applications in the context of deep geothermal projects. Other applications of HF involve the preconditioning of ore bodies with low fracture density (e.g. to induced block caving; Jeffrey et al., 2013; van As and Jeffrey, 2000) and various applied industrial projects, for which a detailed understanding of the stress state is needed (e.g. to optimize the design of an underground facility or pressure tunnels in hydropower; Haimson and Cornet, 2003; Hubbert and Willis, 1957). Furthermore, fluid driven fracturing also occurs naturally, for instance in kilometer-long dikes that transfer magma from deep underground chambers to the Earth's surface or as sills between two older horizontal layers (Lister and Kerr, 1991; Rubin, 1995; Spence and Sharp, 1985).

The context of our study is the exploitation of deep geothermal energy using an approach known as Enhanced Geothermal Systems (EGS). A central aspect of the EGS technology are stimulation operations to develop the reservoir permeability prior to heat exploitation, because the crustal permeability is generally insufficient at depth (Manning and Ingebritsen, 1999). Stimulation approaches include hydraulic stimulation, thermal stimulation and chemical stimulation. For hydraulic stimulation, two prevalent processes for permeability creation can be distinguished: 1) hydraulic fracturing (HF) as the initiation and propagation of tensile fractures and 2) hydraulic shearing (HS), i.e., the reactivation of pre-existing fractures that support shear stress, which promotes shear failure with associated irreversible dilation. These end-members are not mutually exclusive and a combination is possible: McClure and Horne (2014) suggested combined hydraulic stimulation mechanisms such as primary hydraulic fracturing with shear stimulation leak off or mixed-mechanism stimulation. Stimulations in the context of EGS often take place along open borehole sections of several hundred meters (Brown et al., 2012). Such stimulation treatments are usually controlled by the most permeable fractures that are often critically stressed (Barton et al., 1995), and hydraulic shearing becomes the dominant mechanism for permeability creation, at least several tens of meters distance away from the injection interval (Evans et al., 2014). Since this usually does not result in the desired permeability increase, the stimulation intervals can be reduced with packers and thus the injection flow can be controlled zonally. In proposed EGS concepts that includes multi-stage hydraulic stimulation of multiple shorter borehole intervals (e.g. Meier et al., 2015), the initiation and propagation of tensile fractures may become an important mechanism in the near field of the wellbore to connect the wellbore to the pre-existing fracture network and to increase the swept reservoir volume.

Thus, the motivation of this work is to better understand hydraulic fracture initiation and propagation in crystalline rock, as well as the influence of pre-existing geological features and stress heterogeneities on the HF processes. To this end, we performed a series of six extensively monitored HF experiments in the underground laboratory of the Grimsel Test Site in May 2017 (Amann et al., 2018). The first experiments series – six hydraulic shearing tests targeting pre-existing fractures – were performed a few months earlier in February 2017 and are presented by Krietsch et al. (2019a). Both experiment series were accompanied by monitoring of induced seismicity, the results of which are presented by Villiger et al..





## 1.1 Intermediate scale experiments

There are only a few examples of in-situ HF experiments performed in crystalline rock at the deca- to hectometer scale, for example the experiments performed at the Nevada test site (Warpinski, 1985), at Northpark mine (Jeffrey et al., 2009) and in the Aspö Hard Rock Laboratory (López-Comino et al., 2017; Zang et al., 2016). New experiments are being executed or planned at the Homestake mine (EGS collab project; Kneafsey et al., 2018), at the Bedretto Underground Laboratory for Geoenergy research (BULG; Hertrich and Maurer, 2019) and at the Reiche Zeche Underground Laboratory (STIMTEC; Dresen et al., 2019). All experiments differ in terms of the in-situ geological and stress conditions, and also in terms of injection protocols and monitoring concepts.

The Northpark experiment was executed with relatively large injection volumes and rates (max. 16 m$^3$ and 320 l/min), and included a mine-back of the stimulated volume. However, microseismic monitoring was very limited. Thus, seismic events could not be detected in the immediate vicinity of the stimulated fractures. The results of the study illustrate that 1) the stress state is decisive of the overall geometry of the stimulated volume and that 2) the pre-existing fracture network defines the development of the newly created flow paths (Jeffrey et al., 2009). The Nevada test site experiment also included mine-back of the rock volume and highlighted the complexity and tortuosity of the flow path of a hydraulic fracture (Warpinski, 1985; Warren and Smith, 1985). At Aspö, one of the focus was testing the hypothesis that using alternative injection strategies, as for example repeated progressively increasing and cyclic pressurization, may help reducing the number and magnitudes of seismic events. Zimmermann et al. (2019) show results that tend to support this hypothesis, where the cyclic fracturing net pressure seems to lead to a lower seismicity but increases the permeability, although the number of tests is not statistically sufficient. They also show that the maximum magnitude in each fracture phase appears to be correlated to the injected volume in agreement with McGarr (2014) assumption.

## 1.2 Complex hydro-mechanical response

A central aspect of our study focuses on the complex hydromechanical coupling occurring during fracture initiation and propagation, for which fluid pressure and deformation data are key parameters. The pressure response at the injection point has been used to quantify pressure-sensitive permeability changes (Louis et al., 1977) or for stress estimations (Doe and Korbin, 1987; Evans and Meier, 1995; Rutqvist and Stephansson, 1996). Rutqvist (1995) and Rutqvist et al. (1998) combined hydraulic jacking tests with numerical modeling to determine the in-situ normal stiffness of natural fractures or faults in crystalline rock. Hydraulic jacking tests were conducted by a step-wise increase in injection interval pressure. It was concluded that during hydraulic injection into a single fracture in granite, the storativity depends solely on the fracture's normal compliance. Moreover, it depends in turn on the stiffness of both the rock mass and the fracture. Numerical analysis of these tests showed that the flowrate at each pressure step is strongly dependent on the aperture and normal stiffness of the fracture in the vicinity of the injection interval (Rutqvist and Stephansson, 2003). The borehole injection pressure and the injection rate allow the direct estimate of the injectivity of the injection interval, which is again related to the fracture and rock compliance in the stimulated interval.





### 1.3 Seismic response and seismic cloud

Induced seismicity accompanies hydraulic stimulation and can be detrimental to deep geothermal projects when magnitudes of induced earthquakes are perceptible to the public (Ellsworth, 2013; Evans et al., 2012). However, in many industrial projects in the context of both hydrocarbon and heat extraction, including this study, it is an indispensable tool that is used to map the

stimulated fracture system (Maxwell et al., 2010; Niitsuma et al., 1999; Warpinski et al., 2013). When the hydraulic fractures propagate beyond the vicinity of the injection point, they will inevitably interact with natural fractures to some degree. Induced microseismicity (on a small scale also called acoustic emissions) occur as localized brittle-failure processes during high-pressure fluid injection and can be used to approximate the geometry of a single hydraulic fracture or a fracture network. Nolen-Hoeksema and Ruff (2001) proposed three mechanisms that may produce seismicity during hydrofracturing: 1) Tensile

failure at the fracture tip, 2) the stress concentration at the fracture tip causing shear slip along suitably oriented pre-existing fractures and, 3) fluid leak off into pre-existing fractures rising the fracture fluid pressure inducing slip if they support sufficient shear stress. The tensile failure at the tip is typically aseismic or at least radiates a small amount of energy. Mechanism two and three are often seen as the main processes leading to induced seismicity during hydraulic fracturing (e.g. Martínez-Garzón et al., 2013; Rutledge et al., 2004; Warpinski and Branagan, 1989). Thus, induced seismicity does not represent the propagating

fracture itself, but is indicative of the hydraulic fracture propagation as it tracks the propagating fracture. Many experiments on different scales in laboratory and under in-situ conditions showed that the seismicity cloud has a tendency to be oriented normal to the minimum principal stress ($\sigma_3$) direction (Evans et al., 2005; Häring et al., 2008; Hubbert and Willis, 1957; Rutledge et al., 2004). Majer and Doe (1986) concluded from the occurrence rate, spatial and temporal distribution of the microseismic events, that the hydro-fracture growth pattern does not follow an often-assumed single and symmetric fracture path. In fact, the one

"hydraulic fracture" is actually made of multiple fractures.

### 1.4 Our contribution

A detailed characterization of the stress state as well as a geological, hydrological and geophysical characterization of the experimental rock volume took place before the main injection experiments were executed. The in-situ hydraulic fracturing experiment presented here along with detailed pressure, deformation and seismic monitoring were designed to address the

following research questions:

- What is the injection pressure response at the injection interval and at pressure observation intervals in the rock volume?
- What is the rock deformation response to high pressure fluid injection?
- How does permeability increase in response to HF processes?
- How does the microseismic cloud propagate and what is the best description of the fracture geometry? Does the borehole

trace of the hydraulic fracture match the late-time geometry? How does the outcome vary related to the volume of injected fluid?
- How can we describe the stress state during fracture propagation? Does it change?





## 2 Site and rock mass characterization

Our detailed rock mass characterization included geological mapping, hydrological and geophysical testing, as well as in-situ stress measurements.

### 2.1 Site description

The experiment took place in the Grimsel Test Site (GTS), which is operated by the Swiss National Cooperative for the Disposal of Radioactive Waste (NAGRA). The GTS is situated ~450 m below the Western flank of the Haslital valley. The ISC test volume is located in the southern part of the Grimsel Test Site and is accessible by the AU and VE tunnels (Fig. 1). Prior to the ISC experiments, two injection boreholes (INJ1 and INJ2) were drilled from the AU cavern, which were used for the actual high-pressure fluid injections. In addition, another ten boreholes were drilled to access the rock mass for
monitoring purposes: Three boreholes (PRP) were used to monitor fluid pressure, another three boreholes (FBS) were used for strain measurement. The remaining four boreholes (GEO) were used for geophysical measurements during injections such as monitoring the seismic activity or to perform active seismic tests. Three additional boreholes (SBH) were drilled prior to the stimulation experiment during the stress characterization campaign in 2015. A detailed description of the installation of permanent downhole instrumentation systems is provided in a technical description of the ISC Experiment (Doetsch et al.,
2018a).

### 2.2 Geological characterization

The ISC test volume is situated slightly south of the boundary between Central Aare Granite (towards north) and Grimsel Granodiorite (towards south). Alpine deformation and metamorphism overprinted both lithologies, resulting in a pervasive foliation oriented 157/75° (Keusen et al., 1989). See Wenning et al. (2018) for a detailed description of the rock mass in the
GTS vicinity and its deformation history, including a comprehensive reference list. Tunnel mapping in the AU, VE and AU-UP gallery was conducted to identify the main structures. The 15 boreholes drilled as part of the ISC project were logged with an optical televiewer to provide quantitative and qualitative information on the lithology, foliation, and fractures. Based on these data the main pre-existing shear zones have been interpolated within a 3D geological model (Fig. 2b).

The moderately fractured rock mass is crosscut by two sets of shear-zones that differ in terms of deformation history and
orientation. The first set (referred to as S1.0, S1.1, S1.2, and S1.3) included four ductile shear zones that are characterized by a strong increase in the degree of foliation and mylonitization. All four shear zones have an ENE-WSW strike and dip towards SE. These shear zones experienced retrograde brittle deformation and thus contain few discrete brittle fractures. The second set (referred to as S3.1 and S3.2) contains two brittle-ductile shear zones. The S3 brittle-ductile shear-zones consist of a densely fractured zone (>10 fractures per meter) in between two biotite-rich meta-basic dykes.
The three lower hemisphere equal-area pole stereonet with Kamb contour plots presenting each fracture as a pole point within the host rock and the associated shear zone S1 and brittle-ductile shear zone S3 (Fig. 2a). The pole points for the host



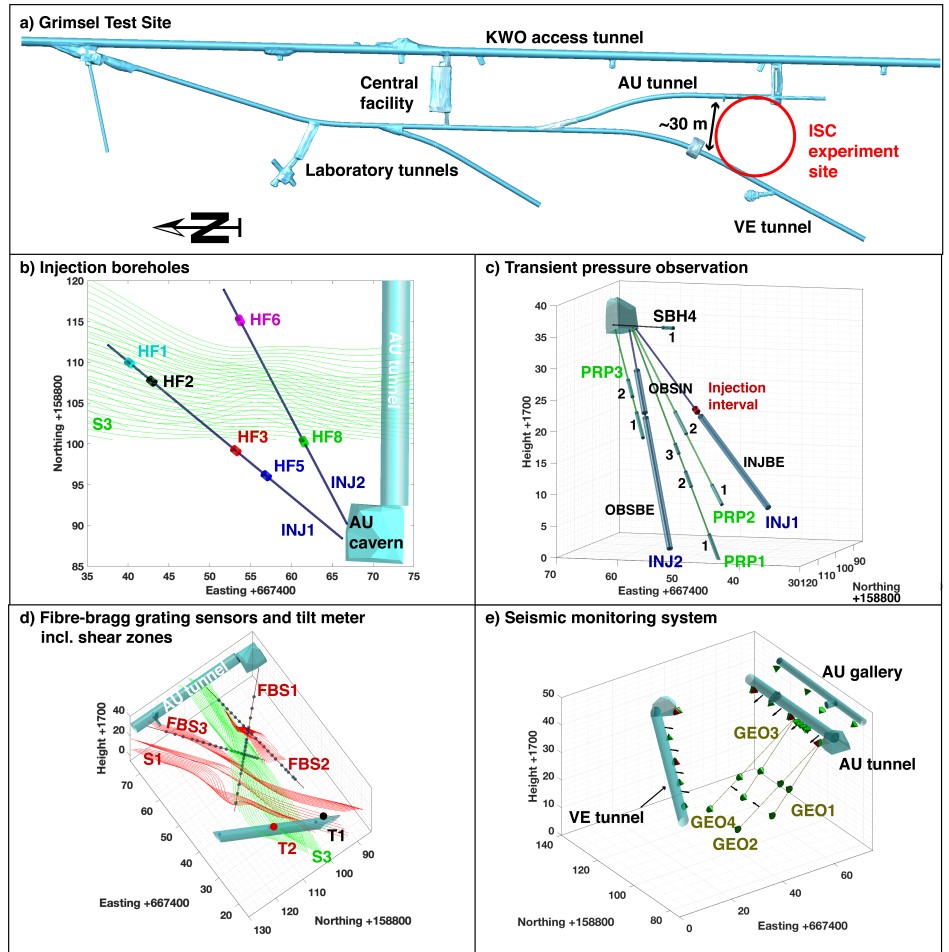

**Figure 1.** a) The ISC experiment site is indicated in the south of the Grimsel test site laboratory facility. b) The AU cavern, the AU tunnel, the S3 shear zone and the injection interval locations in the two injection boreholes are shown and numbered. c) Transient pressure observation intervals in 6 different boreholes. d) Rock mass monitoring systems like Fiber-Bragg Grating (FBG) sensors are indicated by blue circles in the three FBS boreholes and tilt meter are indicated by T1 and T2. The two different shear-zones S1 and S3 are indicated by red respective green. e) The seismic monitoring system consists of accelerometers (red cones) and acoustic emission sensors (green cones) with eight sensors placed in four geophysical monitoring boreholes. Coordinates on figure b) to e) are referenced to the Swiss metric coordinate system (CH1903).

rock and the S1 shear zone indicates a fracture system with a consistent horizontal NNW orientation with a tendency of higher variation in the host rock. The fractures of the ductile-brittle S3 shear zone indicate two different fracture systems.

Figure 2d) shows the fracture frequency, corrected into volumetric fracture density (p32) to account for sampling biases (see Brixel et al., 2019, for the correction), for the two injection boreholes, which indicates a moderately fractured rock mass. The



**Table 1.** Summary of the unperturbed and perturbed stress state

| Stress state | Magnitude | | | Dip/dip direction | | |
|---|---|---|---|---|---|---|
| | $\sigma_1$ [MPa] | $\sigma_2$ [MPa] | $\sigma_3$ [MPa] | $d_1/dd_1$ | $d_2/dd_2$ | $d_3/dd_3$ |
| unperturbed | 14.4 | 10.2 | 8.6 | 104/39° | 259/48° | 004/13° |
| perturbed | 13.1 | 8.2 | 6.5 | 134/14° | 026/50° | 235/36° |

fracture frequency in the INJ boreholes is 0-3 fractures per meter, with higher frequency towards the shear zones. All obtained geological data and the interpolated 3D model were published by Krietsch et al. (2018).

### 2.2.1 Stress characterization

Details of the stress characterization campaign are given by Krietsch et al. (2019b). Impression packers and microseismic monitoring were used to map hydraulic fracture orientation, which revealed consistent E-W, sub-vertical fracture extension. The averaged stress field in relatively unperturbed rock (i.e. with little fracture density) about 8 m from the S3 shear zones (Fig. 1c) and the perturbed stress field are summarized in Table 1. Hence, the minimum principal stress magnitude, measured in the sub-horizontal borehole SBH4 approaching the S3 shear-zone at the borehole bottom, reduces towards the S3 shear zone Krietsch et al. (2019b). The two main changes compared to the perturbed stress field are 1) the stress field is 45° rotated clockwise and 2) the intermediate and minimum stress axis switch place.

Figure 2c) presents the Mohr-Coulomb circle for the unperturbed stress state considering the observed hydrostatic formation pressure of 0.3 MPa. Four different failure limits are presented between 7 and 10 MPa overpressure, assuming a friction coefficient of 0.8 and no cohesion. Fractures and faults mapped from all 15 boreholes are sorted in three categories based on their relation to the main S1 or S3 structures and presented to estimate their criticality due to the fluid pressure increase. Structures favorably oriented for failure will fail with overpressures ranging from 8 to 10 MPa. All the HF experiments are located around the S3 shear zone, which influences the stress field as observed during the stress characterization campaign. To investigate this effect a Mohr-Coulomb circle for the perturbed stress state is presented and the failure limits are indicated for 4 to 7 MPa overpressure. The perturbed stress field would allow shearing of structures above 4.5 MPa overpressure, which is significantly below the observations from the unperturbed stress state. It is not clear, which of the two observed stress states describe the injection into the rock volume approaching the S3 and S1 shear zones best. The experiments executed in borehole SBH4 approaching the S3.1 shear zone indicate the change towards the perturbed stress state.

### 2.2.2 Hydraulic characterization

Multiple field tests were performed to characterize hydraulic conditions at and near the injection borehole pair, including dilution tests, single-hole and cross-hole hydraulic tests. The transport properties of the conductive fractures were characterized using salt and DNA tracer tests (Jalali et al., 2018b). An overview of the hydrogeological baseline conditions for the ISC





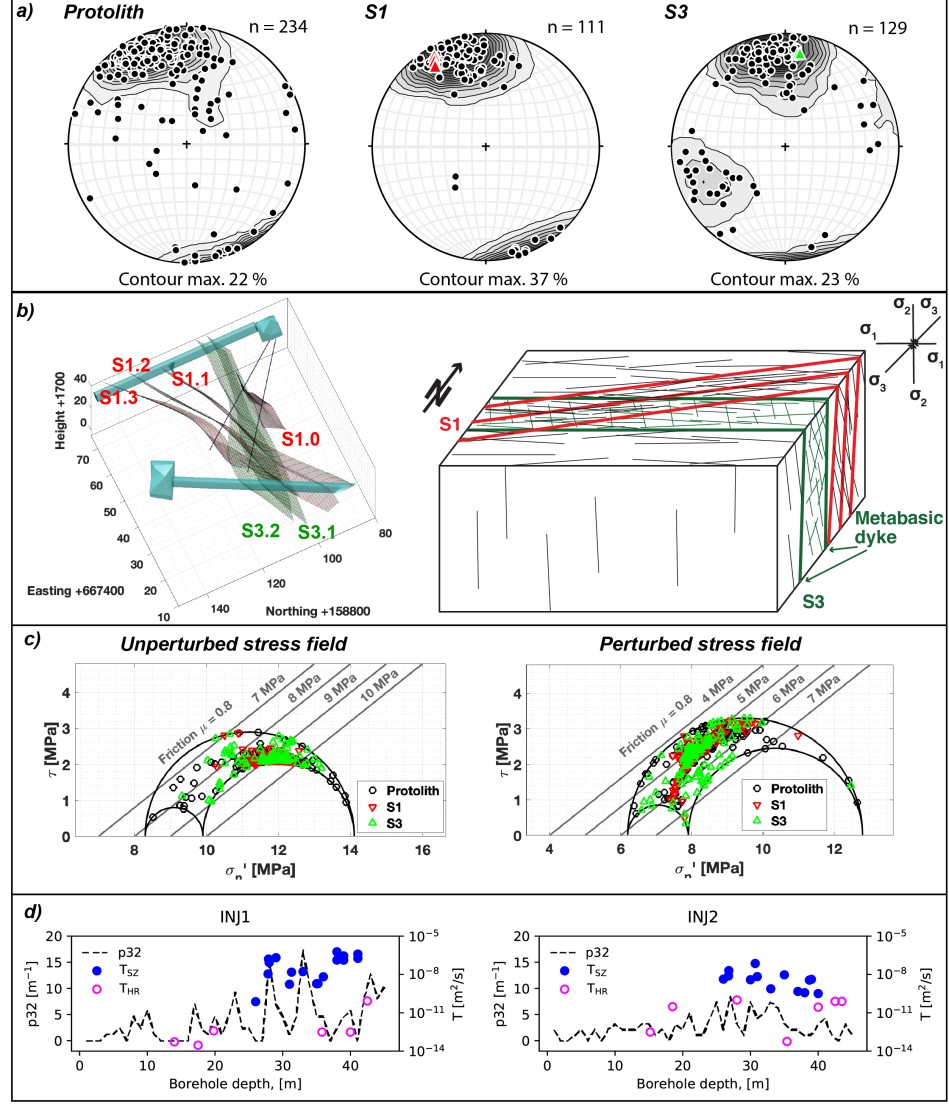

**Figure 2.** a) Kamb contour plots of fracture orientation projected on a lower hemisphere equal-area pole stereonet for the intact rock (host rock), the ductile shear zone S1 and the brittle-ductile shear zone S3. b) Geological model and block model for the ductile shear zones S1 respective the brittle-ductile shear zone S3. The models indicate the different fracture systems associated with the two shear zones. c) Mohr–Coulomb diagram representing the unperturbed and perturbed stress field estimate by Krietsch et al. (2019b) (including hydrostatic pressure of 0.3 MPa). The failure limits assuming a friction coefficient of 0.8. The identified structures from borehole logging in all 15 boreholes are presented to indicate possible failure at a specific overpressure. d) The black dashed line indicates the volumetric fracture intensity calculated over 1 m intervals for both injection boreholes INJ1 and INJ2. The magenta open and blue filled points indicate the position of well tests with the resulting transmissivities from fractured ($T_{SZ}$) and intact rock ($T_{HR}$).





project is presented by Brixel et al. (2019). Hydrogeological conditions prior to the hydraulic fracturing experiment may be summarized as follows:

- The transmissivity of the intact injection intervals (defined here by the absence of visually detected discontinuity on cores and borehole image logs) was estimated through hydraulic pressure pulse tests and range from $10^{-13}$ to $10^{-11}$
$\mathrm{m^2/s}$ (pink circles in Fig. 2d). In injection intervals intersected by shear zones, constant rate injections (CRI) or pulse injections (PI) indicate higher transmissivity values in the order of $10^{-6}$ to $10^{-13}$ $\mathrm{m^2/s}$ (blue dots in Fig. 2d). The geometric average transmissivity of the host rock is estimated to be $10^{-11}$ $\mathrm{m^2/s}$.

- The estimated transmissivities are in good correlation with the fracture intensity in the injection borehole INJ1 (Figure 2d, left).

- Within the brittle fractured zone between the two S3 shear zones an average discharge into the GTS tunnel of ∼60 ml/min was measured prior to the injection experiments.

- Based on the characterization tests conducted, the fractured zone between and along the two S3 metabasic dykes provide the most conductive, natural flow pathways between the two injection boreholes. This observation agrees well with the existence of two different fracture systems in the S3 shear zone: (i) one set following the main NE-SW alpine foliation
orientation and (ii) one set abutting on the two S3 dykes at high angles, which we identified as the alpine tension gashes commonly mapped between dyke swarms throughout the Grimsel Test Site (Figure 2a, right). This rock volume contains open fractures with high transmissivity (i.e. extension fractures), whereby the transport of solutes, salt and DNA tracers between the two injection boreholes show a preferential pathway towards the gallery rather than towards the INJ1 borehole for the case of injection into INJ2 (Jalali et al., 2018b).

## 3    Field setup and monitoring

Hydraulic fracturing experiments conducted within this study were accompanied by an extensive monitoring program including measurements of rock deformation, fracture fluid pressure and microseismicity. In the following sections, we introduce the hydraulic fracturing equipment and the monitoring systems.

### 3.1    Hydraulic fracturing equipment

The HF interval was isolated using a hydraulic double-packer system with a 1 m long pressurization interval. Two different triplex pumps (brand SPECK Pumpen) were used to deliver 1) a pressure up to 30 MPa at a flowrate up to 35 l/min and 2) a flowrate up to 100 l/min at a maximal pressure of 10 MPa. The first pump was used to breakdown the formation and for the first propagation cycle. Then, the pump was switched to reach flowrates up to 100 l/min. A second double-packer system was installed to monitor the fluid pressure response in the monitoring interval in the second injection borehole that was not used
for active stimulation. A data acquisition system recorded the pressure in the open intervals of the INJ boreholes, the flowrate in the injection interval and the packer pressure in the injection and monitoring intervals with a sampling rate of 20 Hz. Fluid pressure was also monitored in the intervals beneath the injection and monitoring intervals with a sampling rate of 1 Hz. The





**Table 2.** Overview of the executed experiment and borehole location, depth of testing interval and observation depth interval.

| Experiment | Date | Start time | End time | Injection borehole | Interval start [m] | Interval end [m] | Observation borehole | Interval start [m] | Interval end [m] |
|---|---|---|---|---|---|---|---|---|---|
| HF1 | 15.05.2017 | 13:30 | 16:30 | INJ1 | 40.0 | 41.0 | INJ2 | 19.6 | 27.3 |
| | 16.05.2017 | 07:30 | 11:00 | | | | | | |
| HF3 | 16.05.2017 | 13:45 | 16:00 | INJ1 | 19.8 | 20.8 | INJ2 | 8.3 | 17.0 |
| HF2 | 17.05.2017 | 08:00 | 11:00 | INJ1 | 35.8 | 36.8 | INJ2 | 8.3 | 17.0 |
| HF5 | 17.05.2017 | 11:30 | 14:00 | INJ1 | 14.0 | 15.0 | INJ2 | 8.3 | 17.0 |
| HF6 | 18.05.2017 | 08:00 | 12:15 | INJ2 | 38.4 | 39.4 | INJ1 | 23.9 | 32.9 |
| HF8 | 18.05.2017 | 13:00 | 15:30 | INJ2 | 15.2 | 16.2 | INJ1 | 23.9 | 32.9 |

injected fluid and the backflow were measured with different flowmeters depending on expected flowrates with a sampling rate of 20 Hz. Figure 1b) presents the position of the six injection intervals along the two injection boreholes INJ1 and INJ2. As a visual aid, we used consistent color throughout the paper to display data from a specific HF experiment. The execution times and the intervals for all experiments are summarized in Table 2.

## 3.2 Monitoring systems

The pressure monitoring system was designed to observe transient pressure response at specific locations to track pressure propagation throughout the rock mass, either through natural fractures or newly created ones. Customized grout packer systems were installed in the PRP monitoring boreholes. The open intervals were packed and separated with hydro-mechanical packers supplement with resin. The uppermost interval was filled with grout to ensure low compressibility of the system. Open hole sections are shown as blue cylinders in Figure 1d. The sections PRP1-1, PRP2-1 and PRP3-1 are positioned within shear zone S1 and all the other intervals are positioned within shear zone S3. The pressure sensors (PAA33-X Keller) were connected to the Solexpert data acquisition system running the Solexpert GM-HF software with a maximum sampling rate of 20 Hz. The possible pressure range of the pressure sensors was 10 MPa with a resolution < 1 kPa. The raw pressure data from HF stimulations are presented in the paper without any filtering.

The rock mass deformation monitoring system consists of 60 fibre-bragg grating (FBG) sensors (Type os3600 by Micron Optics Inc) in the three FBS boreholes. The FBG sensors have a base length of 1 m. 20 FBG sensors were installed along each FBS borehole to characterize the strain field in both intact and fractured rock. The sensors (including strain and temperature) are pre-strained to about 2000 microstrain such that also shortening can be recorded. The sensors were connected to an interrogator of type si255 (Hyperion Platform by Micron Optics Inc.) that can record with a sampling rate of 1 kHz, an accuracy of 0.85 microstrains and wavelength repeatability of 0.1 microstrains. The strain data presented here is not temperature corrected as it is not required for our isothermal injections. Extensional strain is negative.

Two tiltmeters (Type A711-2 by Jewell Instruments) were installed in the VE tunnel to characterize the deformation with respect to the stimulation volume. The location of each tiltmeter is presented in Figure 1c. The tiltmeters measure the deviation





from horizontal tilt in axial (X) and normal (Y) direction to the tunnel with a resolution of 0.05 μradians after filtering with a 100 Hz low pass filter. The two horizontal tilt axis and the temperature were digitized and recorded by the data acquisition system with a sampling rate of 100 unitHz. The initial value was subtracted to display the change of tilt during the HF experiment. Then, a positive tilt in $x$-axis implies a dip of the tunnel wall towards NNE. A positive dip in the $y$-axis indicates a dip

of the tunnel floor towards WNW.

The seismic monitoring network consists of a total of 26 uncalibrated piezo-electric acoustic emission (AE) sensors (type GMuG Ma-Bls-7-70m) and 5 calibrated accelerometers (type Wilcoxon 736 T) (Figure 1e). The AE sensors have a bandwidth of 1 to 100 kHz and their highest sensitivity at 70 kHz. Eight of the AE receivers were deployed in four geophysical boreholes (GEO) in close proximity ($3 - 25$ m) to the injection intervals. The accelerometers have a bandwidth of 5 to 25'000 Hz with a

sensitivity of 100 mV/g. The seismic data were recorded continuously throughout the experiments at a sampling rate of 200 kHz, using a 32-channel acquisition system. AE and accelerometer receiver signals were high-pass hardware filtered at 1 kHz and 50 Hz, respectively.

Based on the picked P-wave onsets, the seismic event locations were calculated using a homogeneous but transversely isotropic P-wave velocity model. For more details on the seismic monitoring and event localization, see Doetsch et al. (2018b);

Gischig et al. (2018); Villiger et al.. The experimental summary cards in the supplementary material (Fig. S8 to S13) present the located seismic events and their radial distance to the injection interval. Time synchronization of all data acquisition units used for the HF injection tests was ensured using a network time protocol server.

## 4    Overview of the HF experiments

### 4.1    Injection protocol

The HF injection experiment was executed between May 15. and May 18., 2017. The injection protocol showing the injected flowrate (blue line) and the injection pressure (red line) for each HF experiment are presented in Figure 3. The grey shaded regions in the plots correspond to the phases of fluid injection. Prior to the first injection phase of each experiment, a pulse injection was executed to test interval integrity and packer sealing. The actual hydraulic fracturing experiment started with a rate-controlled fluid injection at approximately 5 l/min for ten seconds to initiate a hydraulic fracture. This formation break-

down cycle (or short: frac cycle) is indicated by the letter F in the injection protocols and consists of the fluid-injection, the pressure observation during shut-in time and fluid recovery during bleed-off time.

The two following refrac cycles RF1 and RF2 had the aim to propagate the hydraulic fracture. During these cycles, we used either water with a viscosity of 1 cP ($10^{-3}$ Pa.s) (Fig. 3 a-d, experiments HF1, HF2, and HF3) or a shear-thinning fluid (xanthan-salt-water mixture or XSW) with a viscosity of $\sim$35 cPs (Figure 3 e-g, HF5, HF6 and HF8). The refrac cycle RF1

starts with a flowrate of 5 l/min which was progressively increased to 10 and 20 l/min. Each step lasted for approximately 2 minutes or less. For the experiment with water, we performed a cyclic injection during the 20 l/min step. The cyclic injection consisted of a sinusoidal variation of flowrate with a period of 2.5 to 20 s and an amplitude of $\pm$ 15 l/min. Following the 20





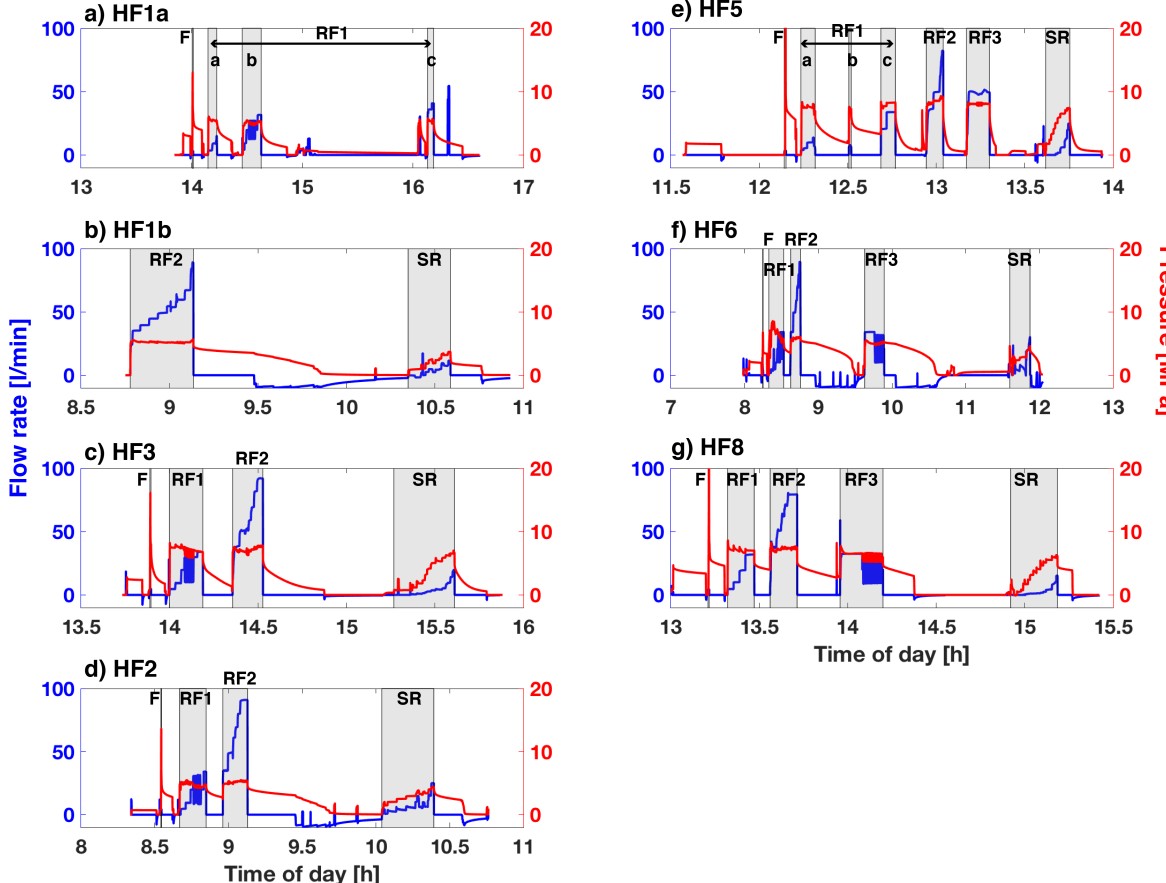

**Figure 3.** The injection protocol of the different HF experiments is presented showing the flowrate (blue line) and the injection pressure (red line). The grey shaded sections correspond to fluid injection into the interval. Experiment HF1 was divided into two protocols due to water supply problems during the first day. The protocols a-g) follow the temporal execution in the field. For the experiments a-d) the injection fluid was water during the fracture propagation cycles (RF1-RF2) and for the experiments e-g) the injection fluid was xanthan-salt-water mixture.

l/min injection step we increased to 35 l/min for 3 minutes. Then, the system was shut-in to change to the bigger pump and injection was resumed without bleed-off with a second propagation cycle called RF2.

The refrac cycle RF2 begins with a rapid increase in the flowrate to 35 l/min and then increased to 50 l/min. Both flow steps were maintained for 2 min. Afterward, each step was held for 1 min starting from 60 l/min and going up to 70, 80, 90 and 100 l/min. The system was then shut-in for several minutes to half an hour to observe the hydro-mechanical response in the system. Finally, the system was opened to allow bleed-off. The fluid recovery rate was monitored. For the HF experiments with XSW fluid (HF5, HF6, and HF8), we added a third refrac cycle RF3 with clean water with the aim of flushing out the XSW fluid.





The water was injected at 35 l/min. This cycle was also an opportunity to test different cyclic injection schemes. The fluid injection was again followed by shut-in and bleed-off phases. For all experiments, the last cycle was a pressure-controlled step test (SR) to evaluate the post-stimulation injectivity of the created hydraulic fracture and to estimate the stress acting normal to the hydraulic fracture (jacking pressure) based on Doe and Korbin (1987). For the executed injection protocols, the following remarks are noted:

- Logistic problem affected the execution of HF1 experiment. The issue was an insufficient water supply. This led to several repetitions of the refrac cycle RF1. The second refrac cycle (RF2) was then executed a day later with a new installed water-supply pump delivering the necessary flowrates. Furthermore, the seismic monitoring system recorded an increased quantity of electronic interferences due to a faulty shielding of the power line between the frequency control unit and pump motor. Therefore, a seismic evaluation was not possible.
- At a flowrate of 5 l/min during the first refrac cycle of experiment HF5, a short-cut occurred to one of the open seismic monitoring boreholes (GEO1, Fig. 1e). For this reason, we had to interrupt RF1 and resume multiple times. This is why RF1 is subdivided in part a, b, c on Figure 3. Thereafter, the pump was changed to allow flowrates above 35 l/min, but the flowrate was limited to a maximum of 80 l/min and short duration due to the short-cut to the GEO1 borehole. The flushing cycle RF3 was executed with a flowrate of 50 l/min.
- The low magnitude of breakdown pressure during the frac cycle of HF6 is an indication for a pre-existing sealed fracture in the open interval. The stimulation interval was mistakenly placed 3 meters further down in the borehole at a pre-existing fracture.
- During all refrac cycles, we never exceeded an injection pressure of 10 MPa.
- Experiment HF3, HF5, and HF8 show a similar fast pressure decay during the shut-in time for the refrac cycles. The same experiments show very small fluid recovery (fluid recovery is shown as negative flowrate in Fig. 3) after the refrac cycle RF2 compared to the experiments HF1, HF2, and HF6.

### 4.2 Diagnostic injection parameters

Table 3 summarizes key observations from the injection protocol (labeling after to Fig. 3). The measurement of breakdown pressure, fracture reopening and instantaneous shut-in pressure (ISIP) followed the ISRM standard presented by Haimson and Cornet (2003). The breakdown pressure represents the peak pressure during injection cycle F. The instantaneous shut-in pressure (ISIP) was obtained during each cycle using the tangent method, i.e. the departure from a linear pressure decrease vs. time occurring right after shut-in (Amadei and Stephansson, 1997). The apparent re-opening pressure ($P_r$) was picked when the pressure change-time step-relationship starts being non-linear (Bredehoeft et al., 1976). The jacking pressure was measured during the pressure-controlled step test SR (more in the supplementary information S1). The cumulative injected water corresponds to the injected volume, $V_i$, indicated for each cycle and the entire experiment. The backflow was measured at the injection interval during venting. Thus, the fluid recovery $V_r$ is only the recovery from the injection interval, however, fluid also escaped from two of the monitoring borehole (GEO) during HF5 and HF8 experiments and from the fractured zones during all experiments. These outflows were also monitored to validate the overall fluid balance over each experiment. Cycle





**Table 3.** Overview of fracture breakdown pressure ($P_c$) and fracture reopening pressure ($P_r$) and additional hydraulic test parameters (injected fluid: water (W) or Xanthan-salt-water mixture (XSW); $V_i$, injected volume; $V_r$, recovered volume) and localized AE event numbers.

| Stage No. | Test-Stage | Fluid | $P_c$ & $P_r$ [MPa] | ISIP [MPa] | $V_i$ [l] | $V_r$ [l] | No. of localized AE |
|---|---|---|---|---|---|---|---|
| 1 | HF1-F | W | 14.9 | 6.6 | 1.7 | 0.6 | N/A |
| 2 | HF1-RF1 | W | 5.7 | 4.5 | 447.4 | 18.2 | N/A |
| 3 | HF1-RF2 | W | 4.9 | 4.8 | 1066.9 | 337.2 | N/A |
| 4 | HF1-SR | W | - | 3.0 | 48.4 | 31.0 | N/A |
| | | | | | **1564.4** | **24.7 %** | |
| 5 | HF2-F | W | 13.95 | 5.5 | 2.8 | 2.3 | 10 |
| 6 | HF2-RF1 | W | 4.6 | 5.6 | 170.5 | - | 208 |
| 7 | HF2-RF2 | W | 4.5 | 4.65 | 646.9 | 226.6 | 313 |
| 8 | HF2-SR | W | - | 3.9 | 141.8 | 46.0 | 0 |
| | | | | | **962.0** | **28.6 %** | **531** |
| 9 | HF3-F | W | 16.3 | 7.7 | 1.7 | 1.7 | 0 |
| 10 | HF3-RF1 | W | 8.0 | 5.6 | 214.4 | - | 26 |
| 11 | HF3-RF2 | W | 6.7 | 5.7 | 621.1 | 12.7 | 49 |
| 12 | HF3-SR | W | - | 4.7 | 71.4 | 1.9 | 0 |
| | | | | | **908.6** | **1.8 %** | **75** |
| 13 | HF5-F | W | 20.5 | 7.9 | 1.9 | 1.0 | 6 |
| 14 | HF5-RF1 | XSW | 6.9 | 6.0 | 188.6 | - | 13 |
| 15 | HF5-RF2 | XSW | 6.8 | 6.3 | 274.3 | 0.1 | 0 |
| 16 | HF5-RF3 | W | 7.25 | 5.9 | 376.8 | 0.2 | 0 |
| 17 | HF5-SR | W | - | 5.8 | 32.1 | 0.1 | 0 |
| | | | | | **873.7** | **0.2 %** | **19** |
| 18 | HF6-F | W | (7.0) | 5.9 | 2.1 | 1.2 | 0 |
| 19 | HF6-RF1 | XSW | 6.3 | 4.95 | 171.1 | - | 0 |
| 20 | HF6-RF2 | XSW | 5.3 | 5.5 | 467.4 | 310.8 | 26 |
| 21 | HF6-RF3 | W | 5.3 | 5.2 | 435.6 | 323.5 | 5 |
| 22 | HF6-SR | W | - | 3.8 | 143.6 | 75.6 | 0 |
| | | | | | **1219.8** | **58.3 %** | **31** |
| 23 | HF8-F | W | 21.2 | 7.5 | 2.3 | 0.9 | 3 |
| 24 | HF8-RF1 | XSW | 7.3 | 5.1 | 165.7 | - | 163 |
| 25 | HF8-RF2 | XSW | 6.3 | 5.25 | 561.7 | - | 18 |
| 26 | HF8-RF3 | W | 7.1 | 4.7 | 386.3 | 8.7 | 8 |
| 27 | HF8-SR | W | - | 4.7 | 31.5 | 11.3 | 0 |
| | | | | | **1147.5** | **1.8 %** | **192** |

RF1 for HF1a and HF5 correspond of multiple refrac cycles. Thus, the re-opening pressure and the ISIP were averaged and the fluid injection respective recovery is presented in a cumulative number.

Figure 4a presents the flowrate ($q_{inj}$) vs the interval pressure ($p_{inj}$) at pseudo steady-state for the second refrac cycle RF2 and the pressure-controlled step test for each HF experiment. For any injection step, the injection conditions (either controlled flowrate or constant pressure) were maintained until a stable state was reached, i.e. quasi constant pressure for rate-controlled injections or quasi constant flowrate for pressure-controlled injections, while we acknowledge that true steady state conditions are never reached in practice, we took the latest data point prior starting the next step of our injection. A major observation is that there is a clear difference between HF experiments executed south of the S3 shear zone (HF3, HF5 and HF8) and north of the S3 shear zone (HF1, HF2 and HF6). HF1 and HF2 were located in the S1 shear zone and reached a pressure limiting behavior at an injection pressure of $5.4\,\mathrm{MPa}$ for flowrates larger than $35\,\mathrm{l/min}$ (Figure 4a). During these two experiments, only two propagation cycles (RF1 and RF2) were executed. Considering experiment HF6, the pressure limiting behavior occurred at a higher pressure of $6\,\mathrm{MPa}$.

HF3, HF5, and HF8 were located south of the S3 shear zone. In general, all experiments executed on this side of the S3 shear zone had higher injection pressure. HF3 and HF8 showed similarly, a slight increase after reaching a flowrate of $10\,\mathrm{l/min}$. The



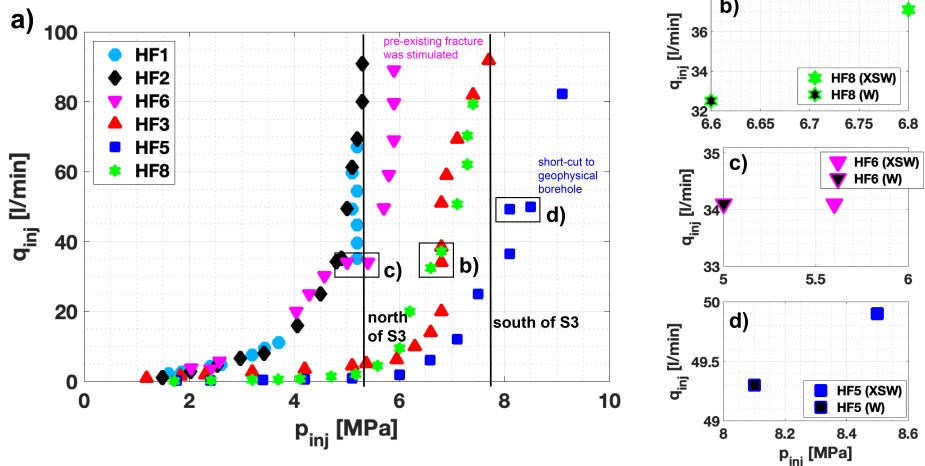

**Figure 4.** a) Presents the injection flowrate $q_{inj}$ vs interval pressure $p_{inj}$ at pseudo steady-state from the second refrac cycle RF2 and pressure-controlled step test SR (similar to 3c) for all HF experiments. (b-d) Highlights the pressure drop when changing from XSW to water with similar flow rate.

injection pressure reached a limiting pressure of 7.8 MPa. The injection fluid for HF3 was water and for HF8 XSW followed by an additional flushing cycle using water. Experiment HF5 showed the highest increase in injection pressure for increasing flowrate with a maximum injection pressure above 9 MPa. The pressure dropped around 0.5 MPa at the same flowrate (35 l/min or 50 l/min) using XSW during the refrac cycle RF2 and the flushing cycle (RF3) using water for experiment HF6 and

HF5. The effect is smaller for HF8 with a pressure drop of only 0.2 MPa. Therefore, we can infer that the viscosity effect (i.e., the change of XSW to water) results in a decrease of injection pressure, but further investigation is necessary to reliably quantify the effect. For the sake of clarity, the flowrate and pressures from the flushing cycle associated with the pressure drop are presented in Figure 4 (b-d).

     The limiting pressure is smaller compared to the hydraulic tensile strength calculated from the difference between breakdown

and reopening pressure (Bredehoeft et al., 1976) ranging between 8.3 and 9.6 MPa. The tensile strength measured by Dutler et al. (2018) ranges between 5.6 and 14.7 MPa for the transversely isotropic host rock. Therefore, the hydraulic tensile strength depends on the orientation of the isotropic plane. The limiting pressure ranges between 5.4 MPa (HF1 and HF2) and 6.8 MPa (HF3 and HF8), and is interpreted as reflecting conditions when the fracture surfaces have fully lifted off near the wellbore. However, it is questionable if the hydraulic fractures still extend (creating new surfaces) at this pressure.

The difference from HF6 to the two other injection experiments (HF1 and HF2) may be related to two reasons: 1) The HF6 interval contains a pre-existing fracture that may not be perpendicular to the minimum principal stress. 2) The injection fluid in HF6 is XSW. Thus, the pressure reaches higher values as pressure dissipation is affected by the high fluid viscosity. During the flushing cycle of this fracture (RF3), a pressure decrease was observed, reflecting a viscosity effect.





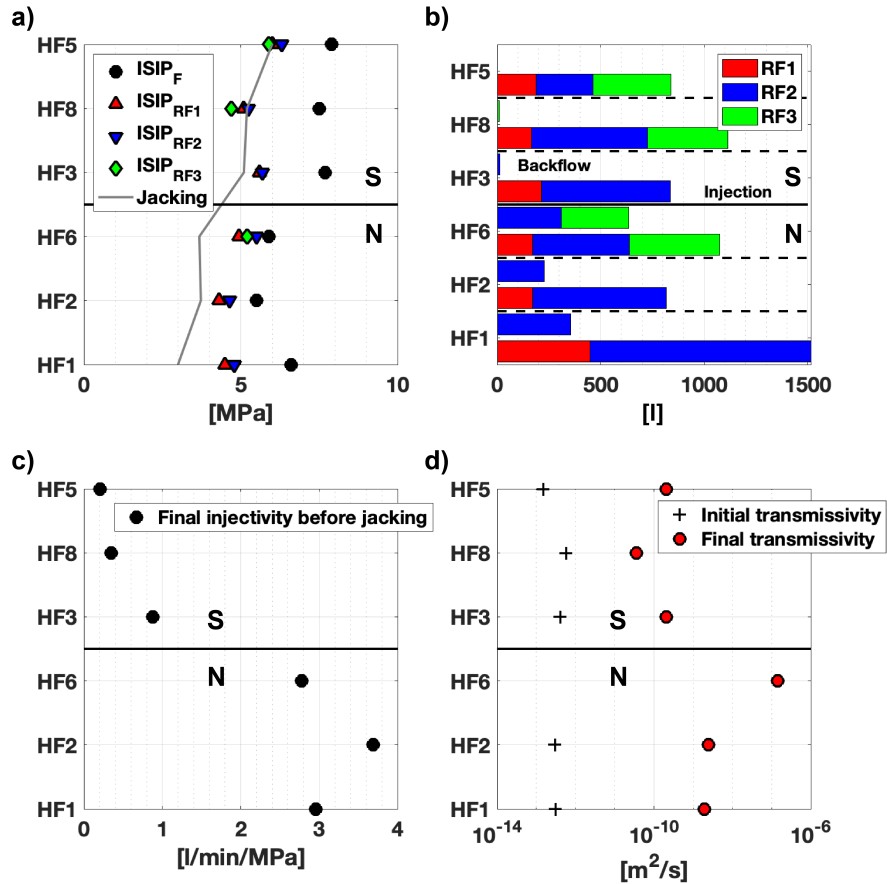

**Figure 5.** The black solid line on all plots indicate the S3 shear-zone, which divide the experiments in north (N) and south (S) with a pseudo-distance ordering the HF experiments on the y-axis from top of the borehole (HF5) to the end (HF1). a) Presents ISIP from frac / refrac cycle and the best estimate of jacking pressure (JP). b) The injected and backflow volume are presented for the 3 main refrac cycles for each experiment. No recovery phase took place after RF1. c) Presents the final matrix injectivity from the pressure-controlled step test. d) The initial transmissivity was measured from pulse injection prior to the HF experiment and the final transmissivity was measured via constant head injection several weeks after the experiment.

Fig. 5 presents four subplots summarizing (a) the ISIP and JP, (b) the injected volume and backflow, (c) the final injectivities before jacking derived from the analyses of the SR cycles and (d) the initial and final transmissivity derived from hydraulic experiments performed prior and after the HF experiment. Fig. 5a compares the ISIP obtained during the frac / refrac cycles with the jacking pressure. The ISIP varies between 5.5 and 7.9 MPa for the frac cycle and decreases with ongoing refracturing. For

5   all experiments, the ISIP stabilizes around 5 MPa and show slightly higher values between 4.5 and 6.5 MPa for experiments executed south of S3 (above the black solid line) and values between 4.0 to 6.0 MPa for experiments executed north of the S3



shear zone. South of the S3 shear zone, the the jacking pressure reaches values between 5.1 and 6.0 MPa, which is comparable with the ISIP from the refrac cycles. In contrast, the jacking pressure north of S3 range between 3.0 and 3.7 MPa for the experiments executed. This reflects 1 to 2 MPa smaller values than using the ISIP from the refrac cycle (Fig. 5a).

The injection volume and the recovery volume from the injection intervals for the main fracture propagation cycles RF1/RF2

and the flushing cycle RF3 are presented in Fig. 5b. The injection volume for each cycle is similar for all experiments except HF1. For HF1 the injection protocol was stopped after refrac RF1 and continued the following day. After the first propagation cycle, no fluid recovery took place. Both, the second fracture propagation cycle RF2 and the flushing cycle RF3 do show minor or no backflow from the injection interval for the experiments executed south of S3 (> 2%). In contrast, for the experiments executed north of S3, the recovered fluid from the injection point reaches values between 15.0% and 23.5%. Experiment HF6

reached the largest volume recovery of all experiments: during the flushing cycle RF3 a recovery of 74.3% was observed. The final injectivity before jacking is presented in Figure 5c and ranges between 2.77 and 3.69 l/min/MPa north of S3 and between 0.21 and 0.88 l/min/MPa south of S3. Approaching the S3 shear zone from the south, injectivity values increase. The experiments located further down in the borehole show highest injectivity values, which correlates directly with an increase of fracture density in the shear zone S1.

**4.3   Transmissivity values from pre- and post-HF hydraulic tests**

The change in transmissivity at the injection interval was investigated by packer testing before and after the HF experiment in borehole INJ1 and INJ2. Prior to the HF experiment, pulse injection (PI) and after the HF experiment, constant head injection (CHI) tests were performed. To estimate the equivalent hydraulic parameters (transmissivity and storativity), the PI tests were inverted with n-dimensional Statistical Inverse Graphical Hydraulic Test Simulator, nSIGHTS (Roberts, 2006) and the CHI

tests were analyzed using the Jacob and Lohman (1952) solution. For both methods the radius of influence is different and therefore the numbers are only an indication of permeability enhancement and not for direct comparison. The initial magnitude of local transmissivity estimates ranges between $10^{-13}-8*10^{-13}$ m$^2$/s. Final transmissivities posterior to the HF experiment, from CHI tests reach values between $1.9*10^{-9}-3.6*10^{-11}$ m$^2$/s. HF6 is an exception as it took place at a pre-existing fracture, for which transmissivity was not measured before the HF experiment. The final transmissivity for HF6 is highest in magnitude

for all HF experiments with $1.5*10^{-7}$ m$^2$/s. There is a trend of one magnitude higher transmissivities after HF for experiments on the northern side of the S3 shear zone (more in the supplementary information S2).

**4.4   Borehole fracture trace**

Prior to the HF experiments, analysis of drill cores, optical and acoustic televiewer have been carried out to select appropriate test intervals. Suitable test intervals for hydraulic fracturing were selected if no pre-existing fractures were visible. Figure 6

presents the amplitude log from the acoustic borehole televiewer for the six HF intervals for pre- and post-testing. The travel-time log is not presented as no changes were recognized. The blue highlighted section in the center indicates the injection interval, which does not show any fractures prior to testing, except for HF6. The areas below and above the blue section show the location of the straddle packer of 1 m length. The pre-existing fractures are indicated in green and located at the straddle-





**Figure 6.** Pre- and post-testing acoustic borehole televiewer (ATV) logs of the borehole INJ1 and INJ2. The blue highlighted sections in the center indicate the open intervals which do not show any fractures prior to testing except for HF6, which was stimulated at the wrong position. The pre-existing fractures are indicated with green arrows. The areas below and above the blue section show the location of the straddle packers. Five of six experiments show new features in the post-testing images which indicates an induced fracture.

packer emplacement. The new fracture trace is indicated by a red arrow. The dip and dip direction were determined by fitting a sinusoidal trace using the WellCAD software by Advance Logic Technology. Based on the probe orientation measurements, the true orientations of the induced fractures were computed. Travel time data accuracy which allows at best to resolve radius





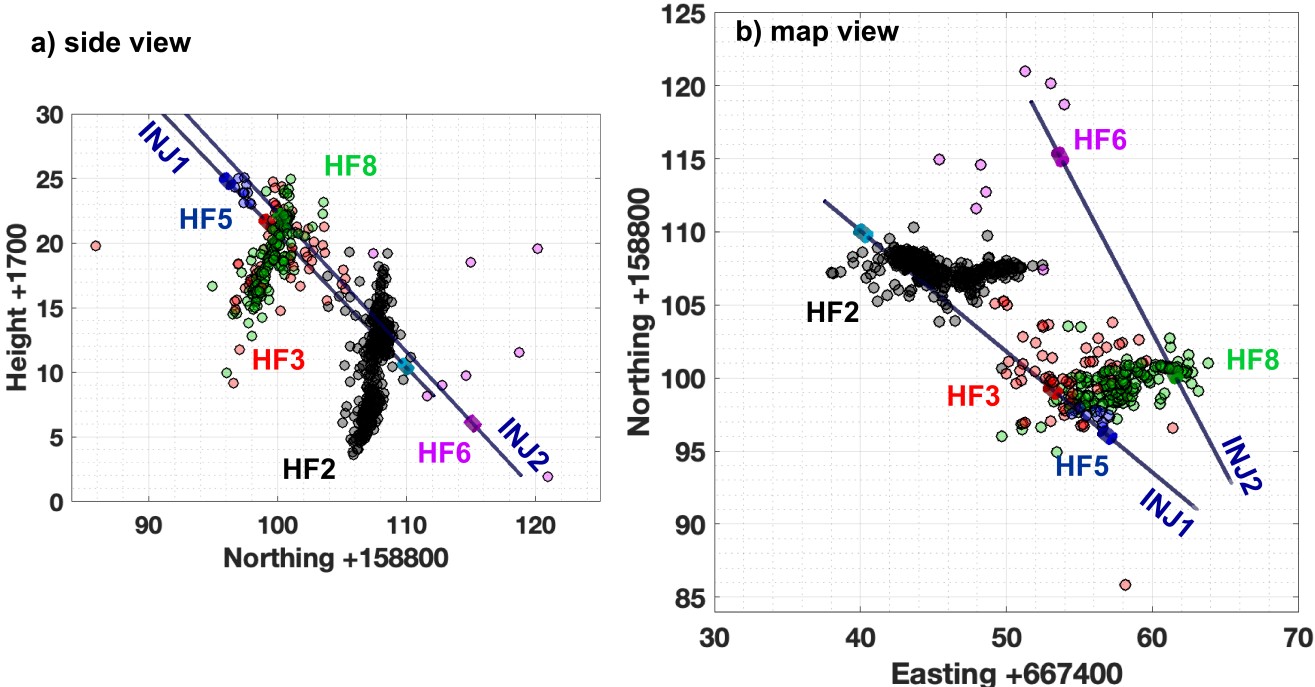

**Figure 7.** Seismicity clouds in side view a) and map view b).

changes of 0.15 mm does not allow to quantify wellbore deformation associated with the presence of the newly created fracture trace, but changes in signal amplitude are sufficient to identify the newly created features during the HF experiments. The borehole traces are either axial to the borehole or have a steep angle to the borehole axis not exceeding 20°. An exception is found in the post-log of HF1, which indicates new en-échelon structures for one side of the fracture trace. We recognized

that for all HF experiments many of the new fractures extend below the straddle-packer and stop at a pre-existing fracture.

### 4.5  Microseismicity

During the HF experiments, we detected in total 6986 microseismic events, from which 730 events were localized. P-wave arrivals were picked manually and located using an absolute location procedure including a joint hypocenter determination (JHD) and a homogeneous, transversely isotropic velocity model (P-wave velocity along the axis of symmetry 5195 m/s,

along the axis of isotropy 4865 m/s). The relative uncertainty of location considering picking uncertainties lies in the range of ±1.5 m, absolute location uncertainties range from below 0.5 m in a depth interval from 15 to 30 m in INJ1 to 1.5 m towards the mouth and bottom of INJ1 and INJ2. Details to the analyzed induced seismicity are presented in Villiger et al.. The number of localized events for each frac / refrac cycle are indicated in the last row of Table 2. In the supplementary information S6, each experiment is presented showing the injection protocol with flowrate and injection pressure, the localized seismic

events for each frac / refrac cycle in plane and profile view and the cumulative injected volume, cumulative backflow and the





distance between injection point and localized seismic events. Figure 7a presents the seismic clouds for five of the six hydraulic fracturing experiments (experiment-wise color-coded).

The number of localized seismic events for experiment HF5 and HF6 is very small. Most seismic activity is observed during experiment HF2 and HF8. Note that the location of the injection intervals with respect to the seismic sensors cannot explain the large difference in detected and located seismic events. During all experiments, the seismic events are located more often below the injection point than above. Also, seismicity propagates predominantly towards east.

A peculiar seismic pattern was observed during HF2 (see Fig. S10 in the supplementary information): seismicity radiates away from the injection point up to a distance of about 5 m during the first part of refrac cycle (RF1), and then propagates to about 10 m away from injection. However, during the second refrac cycle (RF2), seismicity resumes 10 m away from the injection point and develops toward the injection point. This behavior is atypical since migration away from the injection point is typically expected and observed.

Seismicity occurred during all frac / refrac cycles for HF8. Most of the seismic events were observed during the first refrac cycle. The events were located around the borehole with a maximal radial distance of only 10 m. Only three events were located further away during the same cycle. Refrac cycle RF2 and RF3 show less seismic activity compared to RF1, which occurs further away from the injection point.

During the two refrac cycles of HF3, 75 located seismic events occurred. These located seismic events show a dispersed pattern. Most of the seismic events appear below 10 m radial distance from the injection point. Generally, minor seismicity is related with the refrac cycle RF3 and no seismicity is associated with the pressure-controlled step cycle SR for all experiments except HF6.

## 4.6 Fracture geometry

The seismic event locations are shown in a stereographic projection centered on the injection point in Figure 8. Here, each seismic event is represented by the intersection of a unit sphere centered on the middle of the injection interval and a line connecting this middle point and the location of the seismic event. With this method, each seismic event is presented as a point on a lower stereographic projection for experiment HF2 and HF8, respectively (Figure 8). The color of the circle indicates the frac / refrac cycles. In this representation, the location uncertainty of the seismic events impacts strongly the event orientations close to the injection point. A color saturation scheme is used to give less importance to these events: a more intense color is used for events located further away from the injection point and linearly decreasing the color saturation towards the injection points. Events located within a radial distance of 1 m (typical location error for our seismic events) from the injection point are not represented.

In the idealized case of symmetric radial extension of the seismic cloud on a plane from the injection point, the expected pattern on the stereoplot would be a girdle pattern, i.e. all the points will line on a great circle of the stereoplot. In Figure 8 we present the best fit plane through the seismic cloud via girdle pattern for all seismic events (black pole point for HF8) and for two clusters (i.e. RF1 and RF2 for HF2, red and yellow pole point). The two HF experiments presented in Figure 8 do not follow this idealized pattern. In our case, the points tend to cluster that represent a linear structure.



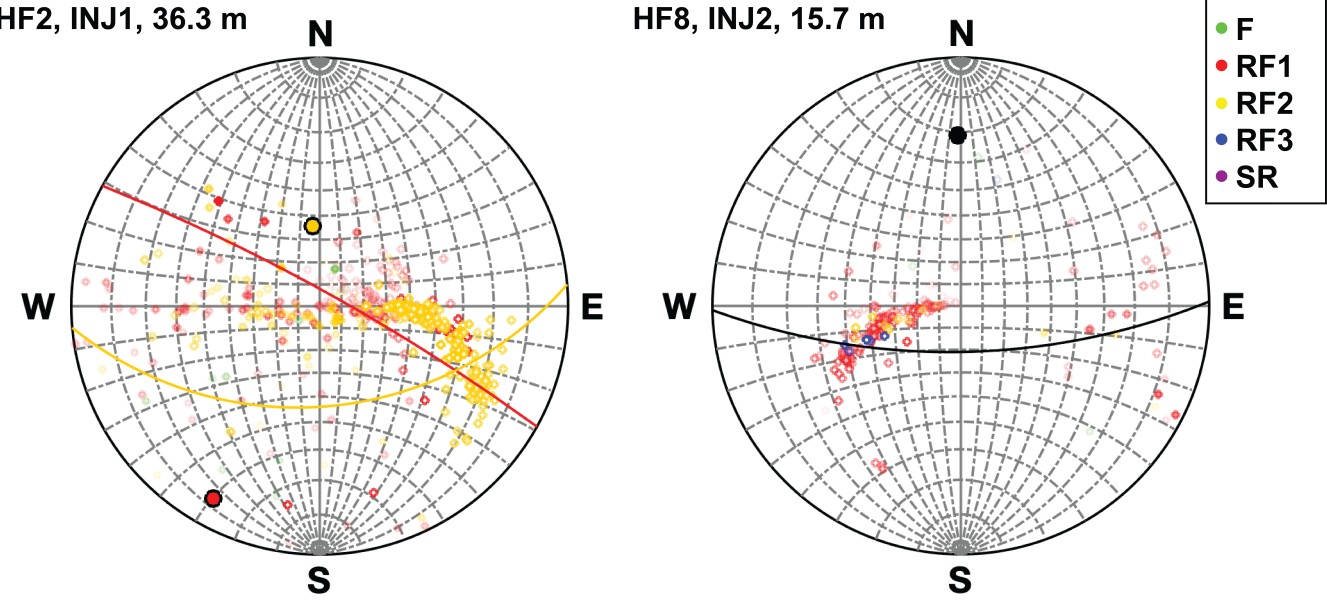

**Figure 8.** The center of the lower stereographic projection corresponds to the injection point and each circle to a seismic event. Circles with a white face color and a different edge color are projection on the lower stereographic net and fully colored circles are projections on the upper stereographic net. The girdle pattern indicates the fitted planes inclusive pole point for different seismic clusters (colored pole point). Considering all seismic events, the girdle pattern and the pole point are black.

For HF2 during RF1 (red), the points tend to distribute on an E-W sub-vertical plane and cluster on a linear structure dipping about 70° to the East. During RF2, the seismicity cloud migrates (yellow) towards a well-defined linear structure with a dip of 25° to the ESE. The mean from the first and second refrac cycle are very consistent dipping 40° to 50° to the East. The seismicity cloud of HF8 is clustered on a linear structure dipping 60° to 70° to the West. Most of the seismic events take place during refrac cycle RF1 and RF2 with more intense color saturation as they are located further away from the injection point. The mean of refrac cycle RF1 to RF3 are very consistent for experiment HF8.

### 4.7 Hydromechanical observations

Two experiments, one executed north of S3 (HF2) and the other one south (HF3) are presented and compared here in terms of deformation and hydraulic pressure distribution in the rock mass. Similar information on the other experiments can be found in the supplementary information S3. For the two experiments presented here, the injection fluid was water and the injection intervals were devoid of fractures prior to stimulation.

The tiltmeter data (Fig. 9 and supplementary information Figure S4) measure the deviation of the VE-tunnel floor from horizontal. The fluid injection is presented by grey shading in the time series of the tilt data. In general, the largest tilt is related to higher injection rates and increasing fluid volume. The magnitude of the tilt axis globally ranges between −4 to 3




**Figure 9.** The time series of tilt meter T1 and T2 are presented for experiment HF2 and HF3. The fluid injection is named and indicated by the grey boxes. The colored points at the end of fluid injection in the time series are presented in a polar plot. The magnitude of the circles of the polar plot are given in radians. The sketch shows the injection points, the orientation of the shear zones and the tilt device location with negative tilt magnitude tunnel parallel $T_{X-}$ respective perpendicular to the tunnel $T_{Y-}$.

microradians. The magnitude of the tilt signals decreases with respect to the injection location and shear zone as follows. Injection executed south of S3 show in general smaller magnitudes than the one executed next to S1, as they are farther away from the tilt meter locations. Also, the response of the tilt signals reacts either instantaneously or delayed depending on the location of injection (distance-controlled response). The experiments next to the shear zone S1 in borehole INJ1 (HF1 and HF2)





show an instantaneous response in the tiltmeter T1 and a delayed response in the y-component of tiltmeter T2. We interpret that the tunnel floor tilts away from the injection volume, which accumulates in the intersection zone of S1 and S3. This zone is located West of the injection point. During the experiment, tiltmeter T1 dips away from shear zone S1. Tiltmeter T2 tilts away from the zone S3 during the first refrac cycle and reorients itself with further injection (RF2) indicating a transient shift
along the S3 shear zone.

Tiltmeter T1 of the experiments south of S3 (HF3 and HF8) indicates tilting of the tunnel floor away compared to the normal of S1.2 shear zone and T2 into the shear zone S3 indicating a small compressive component towards the zone S3.1. Experiments executed south of S3, are expected to connect to the fracture system of the shear zone S3, which acts as a preferential flow path towards the AU-tunnel. The tiltmeter signals are interpreted as secondary deformation fields, where T2 indicates
transient movement of the S1.2 due to dextral shearing along S3. Tiltmeter T1 (located south of S3.2) has a delayed response and indicates movement towards the shear zone S3.1 during the second refrac RF2 cycle. Then, it starts to reorient itself during shut-in and bleed-off time indicating mass diffusion along the preferential flow paths (fracture associated with zone S1.0). Reversible versus irreversible deformation for the tilt meter data will not be discussed in here due to long term changes observed in the time-series (i.e. tides, seasons).

Figure 10 presents selected time series from the FBG strain sensors and the FBG strain data along the FBS boreholes at the end of refrac cycle RF1 and RF2 and the permanent change for the two aforementioned experiments HF2 and HF3. The strain data were set to zero at the beginning of the experiment, as we are interested in the relative changes during an experiment. A positive strain is associated with compression, negative strain represents tension. The permanent strain was either measured 2
hours after the experiment, or at the starting time of the next experiment. The fluid injection is indicated by the grey boxes.

The time series of HF2 show tension during the first refrac cycle in borehole FBS1 at a depth of 31.8 and 33.0 m and compression at a depth of 42.2 m. At the same time when the sensor at 42.2 m starts to release compression, the sensor in FBS2 at 43.3 m starts to show tension and the two other sensors in FBS1 show a decrease in tension. During shut-in, all selected FBG sensors show an immediate decrease in tension except the sensor at 42.2 m in FBS1, which still shows an increase in
tension until the injection interval was bled off to release the pressure. The strain sensors along the borehole FBS1 show the largest magnitude compared to the sensors in the other boreholes. The FBG sensors at 31.8 m and 33.0 m indicate a fast-tensional increase during RF1, linking to the arrival of the fluid front (i.e. the sensor at 33.0 m with one fracture along the sensor base length). The sensor at 31.8 m was placed in intact rock and indicated only compressional signals prior to the HF2 experiment. Therefore, it was hit by the hydraulic fracture during RF1. The highest tensional signals are observed in the S1
shear zone for experiment HF1, HF2, and HF6. Above 35 m the FBG sensors in FBS2, which is oriented parallel to the S3 shear zone and crosses the S1 shear zone at 35 m, indicates compression above 35 m and tension below 35 m. This is directly related to fluid flow through fractures intersecting the base length of the FBG sensors related to the S1 shear zone. The FBG sensors in FBS3 show no signal above 30 m. A small compressional signal is observed between 30 and 35 m and tensional signals below 30 m. The tensional signal is observed in the S3 shear zone, indicating fluid flow towards south. The brittle-ductile shear zone
S3 response differs approaching the deeper volume, where FBS3 intersects the S3 shear zone compared to FBS1 (no response).



**Figure 10.** Selected time series of FBG sensors from borehole FBS1 and FBS2 for experiment HF2 a-c) and HF3 d-f) inclusive sensor location. The fluid injection time is named and indicated by the grey boxes. The depth of the sensors measured from top is indicated in the specific legends from the time series. Positive strain indicates compression.

The time series of HF3 in Figure 10 show two FBG sensors at 22.35 m in FBS1 and at 20.0 m in FBS2 with very high transient peak strain. The time series of the FBG sensor at 20.0 m in FBS2 from HF3 indicate the largest transient peak strain





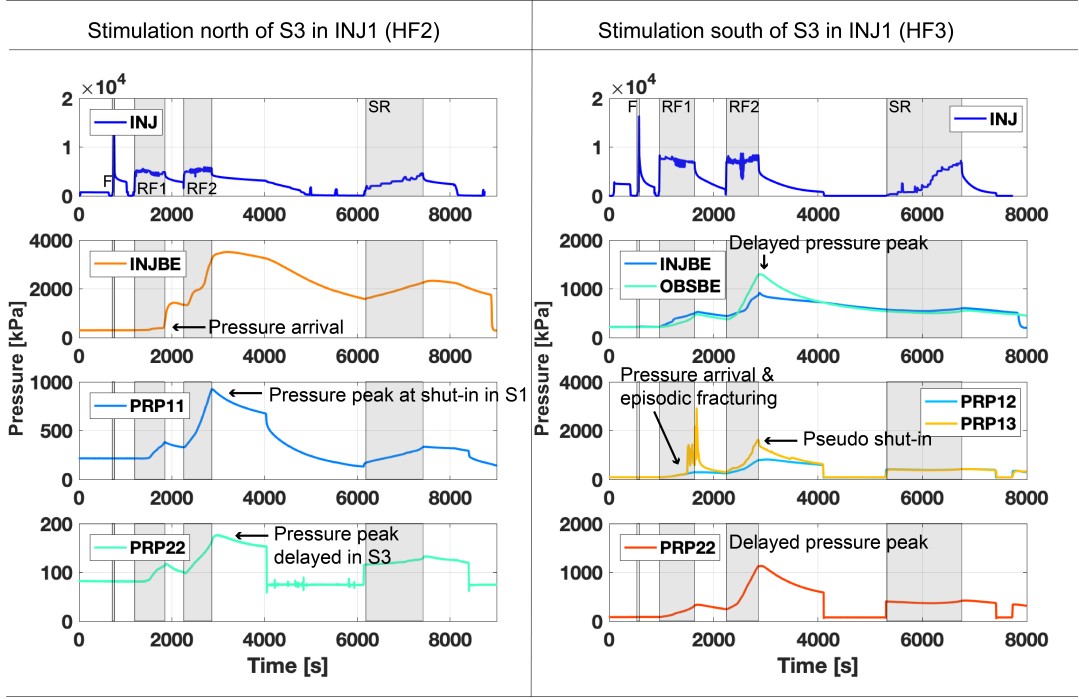

**Figure 11.** Selected time series of pressure response are presented for experiment HF2 and HF3. The fluid injection is named and indicated by the grey boxes.

during all HF experiments. Most of the observed peak strain is reversible and 50 µε are irreversible (permanent) strain. We assume that this tensional reversal strain is directly linked to the fluid pressure. The sensor at 23.35 m in FBS1 is located at the metabasic dyke. This sensor shows compression, whereas the sensors in between the two metabasic dykes indicate tension (e.g., the sensors at 24.3 and 24.8 m). The high transient strain signals (i.e. FBS1 22.35 m and FBS2 20.0 m) indicate major

5 flow paths and are only observed for experiments executed south of S3. During the first refrac cycle, the two aforementioned sensors show small compression before they reverse to tension. This can be interpreted as propagating fluid front that reaches the brittle-ductile shear zone S3, which drains the injection fluid towards the AU-tunnel. The behavior of sensor at 24.0 m in FBS2 is similar but delayed. It shows compression at the beginning which changes to compression at the end of refrac cycle RF1. At the same time, the sensor at 20.0 m in FBS2 shows a sharp drop and starts to compress. In FBS2 only 3 FBGs observe

10 tensional strain, while all the other sensors are compressed, with the highest compression towards the extensional section. Along the borehole FBS3, the sensor associated with S1 shears and intact rock show compression. Note, that the most of the tensional strain signals are observed on sensors associated with the brittle-ductile shear zone S3 (i.e. FBS1 23-30 m and FBS3 > 33 m). Large permanent strain changes are not observed (changes are in a range of 20 and −30 µε).





Selected time series from pressure observation intervals in the injection and observation boreholes are presented in Figure 11 for the hydraulic fracturing experiment HF2 and HF3 (for the other experiments see supplementary information). The injection pressures (INJ) is presented on top including the grey boxes, which indicates fluid injection.

For the stimulations north of S3 (HF2) the time series from the interval below the injection (INJBE) shows a pressure arrival
directly at the end of the first refrac cycle RF1. The peak of the pressure response in PRP1-1 is at the end of refrac cycle RF2. The interval PRP2-2 is located in the shear zone S3 and the injection location is next to the S1 zone. It is expected that the fractures related to the S1 zone are connected first and therefore the interval PRP2-2 located in the S3 shear zone has a delayed pressure peak. The interval INJBE is located next to the injection interval (in shear zone S1.2 and S1.3), showing pressure arrival after cycle RF1. The nearly instantaneous pressure response indicates a new complex flow path connection to
the injection interval below. In addition, the largest amplitude is observed, compared to all other observation points and the maximum peak after shut-in is delayed comparing interval PRP1-1. This indicates a longer fluid path reaching interval INJBE than PRP1-1.

The HF3 injection experiment triggered a strong signal during the first refrac cycle in the observation interval PRP1-3: it indicates an abrupt pressure arrival and an episodic pressure change during and after pump shut-off. At the end of cycle RF2,
pseudo shut-in is observed at the interval INJBE and PRP1-3: i.e. both pressure curves drop after pump shut-off. The pseudo shut-in in the observation interval near the injection interval is a mechanical response starting from the injection interval due to instantaneous pressure loss. The observation interval OBSBE and PRP2-2 respond with a delayed pressure peak, where the two intervals are located further away than the two intervals (INJBE and PRP1-3) showing pseudo shut-in.

## 5  Comparison between stress characterization and HF experiments

In the following section, we compare the hydraulic fracturing experiment with several small-volume ($\sim$10 l) hydraulic fractures from the stress characterization phase (hereinafter called: minifracs (MF)). The minifracs were accompanied by micro-seismic monitoring to investigate the initiation and propagation of the new induced fractures. A detailed description of the stress characterization and the minifracs has been published by Gischig et al. (2018); Jalali et al. (2018b); Krietsch et al. (2019a). In the separate experimental summary cards, each minifrac experiment from SBH3 and SBH4 boreholes is presented in a similar
way than the HF experiments. A summary of the injection pressure and seismic characteristics of all MF experiments (SBH1, SBH3 and SBH4) can be found in the supplementary information S4 and S6 inclusive an overview of the injection interval locations.

### 5.1  Injection pressure observations

Figure 12 summarizes the breakdown pressure and the ISIP from all MF and HF experiments. Breakdown pressure and ISIP
decreases towards the S3 shear zone. The highest decrease is observed in borehole SBH4 approaching the S3 shear zone at the end of the borehole. The experiments executed in borehole SBH3 (MF1 – MF3) show the highest breakdown pressure between 23.4 and 26.1 MPa. The shut-in pressures range from 8.1 to 9.1 MPa. The experiments in SBH1 have smaller breakdown





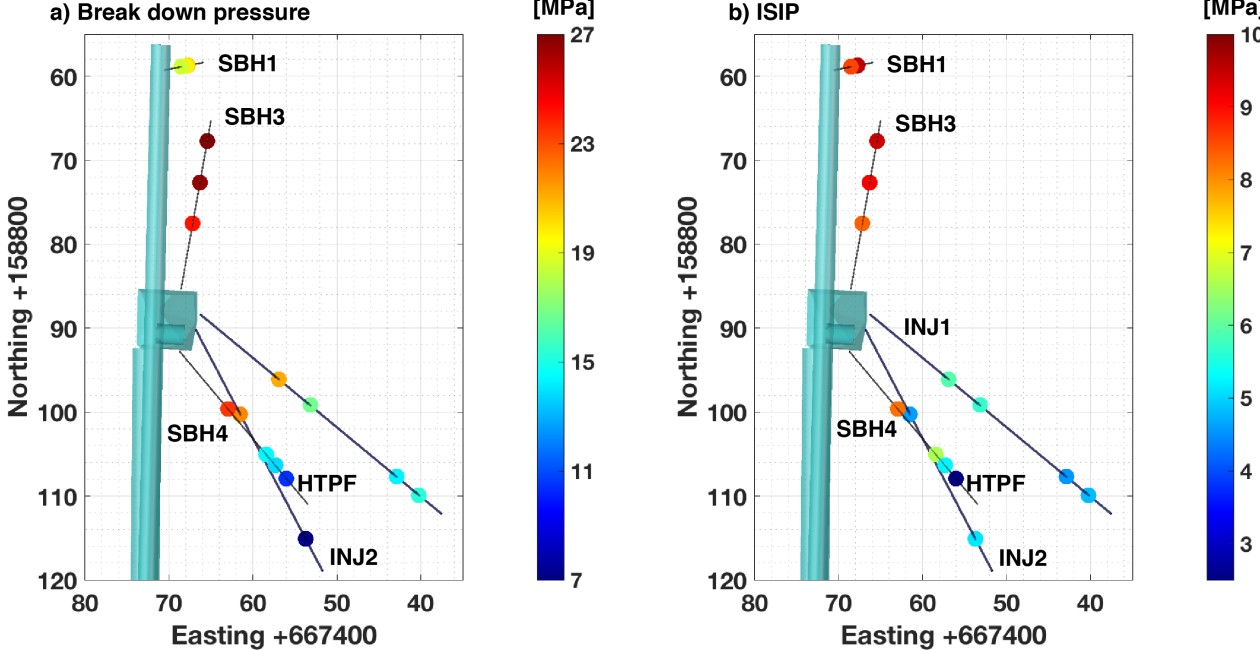

**Figure 12.** Location of HF and MF experiments, viewed towards west with a color-coded for a) breakdown pressure and b) instantaneous shut-in pressure (ISIP).

magnitudes ranging from 13.8 to 19.8 MPa, where the ISIP with values between 8.3 and 9.6 MPa is comparable with the minifracs executed in SBH3. The minifracs from SBH4 (MF4, MF6-MF7) start with a breakdown pressure of 22.7 MPa, which decreases towards the S3 shear zone reaching a value of only 13.5 MPa for the minifrac approaching the S3 shear zone. The ISIP shows a similar decrease from 8.0 to 5.3 MPa. The additional hydraulic test on the pre-existing fracture (HTPF) executed in the S3 shear zone indicates a joint breakdown pressure of 10.3 MPa and a jacking pressure of 2.8 MPa. The breakdown pressure of our larger-scale HF experiments does also show a decrease towards the S3 and S1 shear zones, but the change in magnitude is significantly smaller with magnitudes around 21.2 to 16.3 MPa south of S3 and 14.9 to 13.9 MPa north of S3. The ISIP was measured during cycle RF2 for the experiments using only water and during cycle RF3 for the experiments using XSW. Generally, we find that the minifracs executed in SBH1 and SBH3, i.e. away from the shear zone, have larger breakdown pressures and ISIP than the experiments performed in SHB4, INJ1, and INJ2 that are closer to the S1 and S3 shear zones. In addition, the experiments performed in the more fractured rock mass in the vicinity of the shear zone show larger variability reflecting stress heterogeneity.

## 5.2 Fracture geometry

In the following, we compare the number of localized seismic events between the HF and MF experiments. Then we compare the geophysical borehole logging/impression resulting in fracture traces at the wellbore with best fitting planes through the





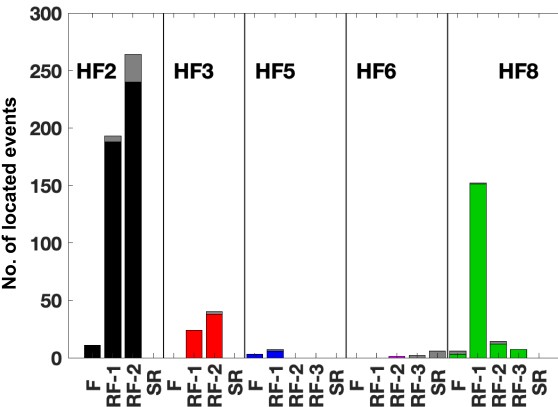
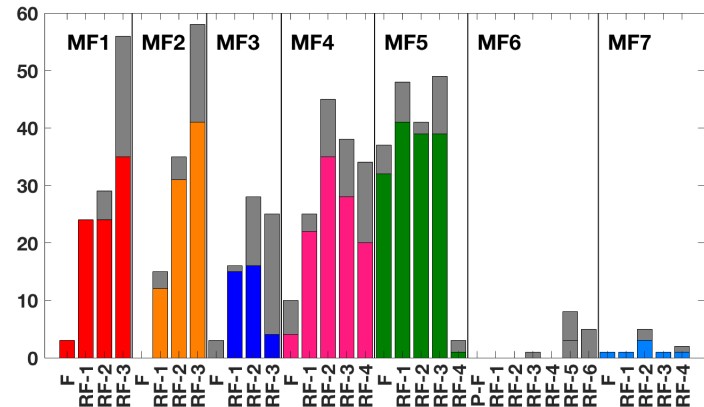

**Figure 13.** The hydraulic fracturing (HF) and minifracs (MF) from the stress characterization phase are presented in an event versus frac / refrac histogram. The color in the histograms corresponds to the number of events during injection. The grey bar on top of the histograms indicate the seismic events after injection shut-in.

microseismic clouds giving information on the fracture orientation away from the wellbores. One should be aware that the seismic monitoring array differed between the HF and MF experiments, which results in different network sensitivities and location accuracies. The number of located microseismic events for all HF experiments and minifracs during each frac (F) / refrac (RF) cycle are presented in Figure 13. All localized events occurring during injection are indicated by colored bars, while the grey bars on top indicate the events after fluid injection (during shut-in and bleed-off time). The following key findings from Figure 14 can be noticed:

- On average, the total number of located seismic events do not differ significantly between the MF and HF experiments, although the variability from experiment to experiment is large. This is somewhat surprising knowing that about 10 l was injected during the MF experiments and 1000 l during the HF experiments.

- The variability of the number of seismic events is particularly obvious when comparing MF6, MF7, HF5, and HF6 (all experiments exhibit less than 50 events) with for example HF2 (531 events). Note that it cannot be ruled out that the different seismic network layouts for HF and MF affect the sensitivity towards detecting lower magnitude events and thus can influence the result.

- However, looking at frequency-magnitude distributions and the corresponding magnitude of completeness analyses presented by Villiger et al. suggests that for the HF injection experiments, not all the variability in detected and located seismic events can be attributed to sensitivity variations of the seismic network. A possible explanation for the low number of located events during MF6 and MF7 is the fact that they were close to the shear-zone S3. During injection experiment HF5 a direct short-cut to a geophysical observation borehole was created. Injection experiment HF6 was executed at the wrong borehole interval, where a pre-existing fracture was stimulated, which was already stimulated before during the hydraulic shearing (HS) experiment.





- The minifracs show a tendency to have relatively more seismic events during shut-in (34.2 %) and bleed-off time compared to the hydraulic fracturing experiments (5.9 %).

- During the MF experiments, typically a small number of events occur during the formation breakdown cycle (F) compare to the refrac (RF) cycles, except for MF4 and MF5. MF5 had an insufficient sealing of the experiment section allowing the fluid to by-pass the packer. Nevertheless, the seismic response is high and the experimental summary cards (supplementary information) indicate that most of the seismic events are located around the borehole up to 4 m away from the injection point, with some minor events located 7 to 10 m away from the injection interval.

- During the HF experiments, most seismicity occurs during the refrac cycles (RF) whereby the injected volume and the flowrate progressively increase. Except for experiment HF6, no seismicity was observed during the pressure-controlled step (SR) test. Prior to this experiment, we opened the valve of the injection interval to drain the fracture system. The small injection volume during the pressure-controlled step test and the small flowrates were not sufficient to re-initiate micro-seismic activity.

The poles to the fracture traces determined at the borehole wall from acoustic televiewer data (see Fig. 5) in the boreholes INJ1 and INJ2 for the HF experiments are presented in Figure 14a and the pole to the fracture traces from the minifracs using impression packer are presented in Figure 14b. We assume, that these traces correspond to the fractures initiated during the frac-cycle (formation breakdown). The fractures during HF experiments presented in Figure 14a are sub-vertical with a N to NE dipping direction. The HF traces are axial or make a small angle to the injection borehole axes. The foliation orientation (337/15°) is indicated by the magenta pole in Figure 14, which is also corresponding to the main brittle fracture set. The minifracs MF8-MF11 were executed in the sub-vertical borehole SBH1. The orientation of the minifracs (Fig. 14b) are primarily aligned with this foliation plane except for MF09 and MF10 that have a similar trace orientation than for the HF experiments. The minifracs MF08 and MF11 orient towards the foliation and the brittle fracture orientation in the host rock (compare Fig. 2a), such that the fracture opened either along the foliation. The orientation of fracture trace MF01 and MF02 from the sub-horizontal borehole SBH3 show a radial fracture initiation, which highlights the dominating control of foliation as such orientation is not the most favorable for initiation at the borehole from a stress concentration viewpoint.

The pole points from the plane fit to the seismic clouds are presented in Figure 14c for the HF experiments and Figure 14d for the minifracs. For both HF and MF, most of the seismic best fit planes are sub-vertical, striking E-W and slightly dipping to the South. Note that some of the plane fits are poorly constraint due to the small number of localized seismic events, i.e. HF3. Also, we noticed a change of orientation for HF2 with injected volume. During the first refrac cycle RF1, the seismic cloud orients itself vertically, striking ESE-WNW. Considering all seismic events including refrac cycle RF2, the seismic cloud orients itself sub-vertical, striking E-W similar to the minifracs. For experiment HF3, two different clusters are possible depending on the seismic events. The first fit indicates a sub-vertical plane striking towards S-N with a misfit of 1.2 m. The second cluster orients itself sub-vertical in E-W. The data presented in Figure 14 are summarized in the supplementary information (S5) with wellbore trace description and best plane fit details.

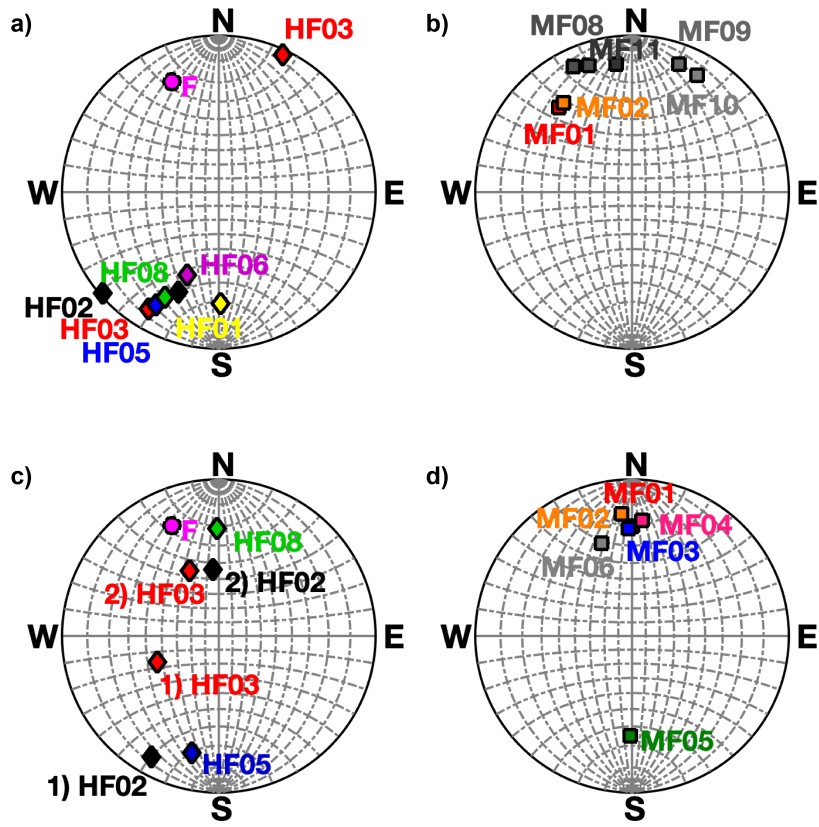

**Figure 14.** The newly created fracture traces from geophysical borehole logging and impression packers are presented in a lower hemisphere stereonet for the HF experiment a) and the MF experiment b). The pole point of the plane fits from the seismic events are indicated in a lower hemisphere stereonet for both the HF experiment c) and the MF experiment d). The foliation is indicated by the magenta circle in panels a) and c).

## 5.3 Fracture propagation

Fracture propagation during the fluid injection can be tracked using the seismic events, which move away from the injection interval. In Fig. 15, all seismic events from the minifracs (MF) are presented with grey circles. The grey circles are absent after 10 l of injected volume as it is the maximum injected volume for these experiments (Figure 15). Further, the circle shows two localizations, one along the 10 m axis and the other increase with increasing injection volume starting at around 2 m for 0.3 l and ending at around 7 m for 10 l. The grey circles at 10 m are associated with the drained and disturbed stress field around the AU-tunnel. On the other hand, the colored circles present the HF experiments. They show a positive correlation between distance and fluid injection. The distance stops to increase after an injection of 60 l and stays around 10 m. The injection of 1





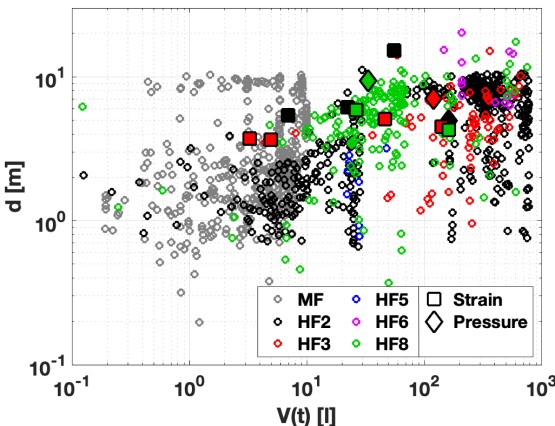

**Figure 15.** The corrected injection volume incl. backflow correction is presented against the distance from the localized seismic events to the midpoint of the injection location. The grey circles present all the localized seismic events from the minifracs without subdividing them. The other circles present different HF experiments. The squares are tracked from FBG sensors or the distributed strain system (DSS) and the diamonds are taken from the transient pressure interval.

m3 fluid into the fracture network results in observed seismic events maximally 20 m away from the injection interval, whereby the 10 l injected during the minifracs show a maximum distance between injection interval and farthest away event of 7 to 10 m.

Instantaneous tensional strain and pressure response are tracked through our experiments to get the geometry of the hy-
draulic fractures. The midpoint of the strain or pressure sensor/interval and the injection point is known and the picked time is calculated for the corresponding corrected injection volume. The fracture distance of HF3 agrees between strain and seismic observation for small injection volume. With increasing injection volume, the seismic events propagate towards 10 m for an injection volume of 700 l. Two strain measurements indicate fracture opening 5 to 6 m away from the injection point and instantaneous pressure increases 7 m away for an injection volume of 50 to 105 l. For experiment HF8, the strain measurements
at 30 l indicate a distance of 6 m from the injection point, where the farthest seismic event is 7 m away from the injection interval. The pressure data point from experiment HF8 has a distance of ∼9 m and respond after 20 l of injection. At the same volume, the seismic cloud indicates a distance to injection of around 8 m. Considering the strain and pressure measurements from experiment HF2, we see that the distance of the seismic events is two times smaller compared to a strain sensor. It is remarkable that the seismic events are restricted to within 10 m distance at high-pressure injection even though we know the
hydraulic fracture is 15 m in length (HF2 strain sensor at 15 m). This agrees well with the observation of Warpinski et al. (2013) that the hydraulic fracture is essentially aseismic and that the stress-induced microseismicity is located at fractures with a significant dimension around the hydraulic fracture.





## 6 Discussion

### 6.1 Hydraulic and mechanical response to hydraulic fracturing

Hydraulically, two different behaviors were observed in experiments performed south of S3 compared to experiments performed north of the S3 shear zone. The differences consisted of lower pressure levels (jacking and ISIP), larger recovered
water volumes, and larger final injectivity for experiments north of S3. Our proposed explanation of this behavior is as follow.

For the experiments HF3 and HF8 (both south of S3), the injected fluid was drained by the densely fractured S3 shear zone towards the AU tunnel, where the S3 shear zone is bounded by two metabasic dykes and therein two subnormal fracture systems are found. The injection related to this geological feature (south of S3), which acts as strong hydraulic boundary condition, therefore, be described as an open system. Thus, no fluid was recovered through the injection borehole for these injection
tests (less than 2%). This configuration also favored direct flow connections reflected by instantaneous responses in pressure monitoring intervals. The tilt signals were rather small as no fluid was accumulated and stored in the fracture system and the FBG-strain sensors show only for specific locations high tensional signals, where in general the fluid circulates through the fractures. Sensors located in the intact rock mass show compressional signals with increasing magnitude towards the sensors with the high tensional signals. A tensional signal on the FBG sensors is predominantly observed in the S3 shear zone if the
injection location was south of S3. This means the FBG sensors indicate fluid path changes during the experiment HF3, as sensors first close and starts to open with a time delay. At the same time, the sensors, which already extend show a stabilization or decrease in negative strain (Fig. 10). For the injection locations north of S3, tensional signals are observed in the S1 zone and either tensional or compressional signals can be observed in S3.

The injected fluid for the experiments executed north of S3 (HF1, HF2, and HF6) was to some extent stored in the fractures
associated with the S1 zone. The fluid recovery from the injection location ranges between 15.0-23.5% for the first two refrac cycles. The flow field is complex and extends most favorably towards the S3 and S1 intersection zone, where the highest fracture density is observed. This is supported by the tilt signal and the FBG-sensors, which have bigger tilt and smaller strain signals than for experiments executed south of S3, where the FBS-boreholes are closer to the injection location and the tiltmeter farther away. Injecting north of S3 (HF2), the FBG sensors indicate tension in the S1 zone along borehole FBS1 and in the S3
shear zone along borehole FBS3, but compression in the S3 shear zone along the FBS1 borehole (Fig. 10). There, a flow path exists between the S1 and S3 shear zone in the lower section of the volume, but the injected fluid volume is either too small or the connectivity is high such that the FBG sensors along borehole FBS1, which is located further above, show a compressive signal.

### 6.2 Borehole trace and fracture tortuosity

The comparison of the fracture trace orientation at the borehole wall observed by acoustic televiewer logging with the orientation of the seismic cloud associated with a given fracture allow assessing fracture rotation, also referred to as tortuosity, in the near field of the borehole. An angular difference of about 30° is typical for most of our experiments, which highlights the uncertainty in determining the stress orientation from direct wellbore information only. For the minifracs, the scatter in the





trace orientation is larger than the seismic cloud data. These experiments were primarily performed in intact rock and affected a small volume of intact rock around the borehole. For the HF experiment, the scatter in the seismic cloud is larger. This is interpreted as reflecting the dominating effect of pre-existing fractures. With larger injected fluid volume, the test volume will inevitably reach some significant features that affect the geometry of the seismic cloud.

The fracture traces observed at the borehole wall mostly extended below the packers. It is unclear if they initiated at the packer location or grew beyond them later during the successive propagation phase of the experiments. Nevertheless, the fracture geometry at the borehole wall is generally complex and its relation to the far field stress orientation, as typically assumed, may be misleading; at least in boreholes which are not aligned with one of the principal stress components.

### 6.3 Seismic response and fracture geometry

A key element to track fracture geometries is microseismic event clouds. Strain response and pressure observations also permit to some extend to track the geometry of hydraulic fractures. These data allow us to show that the geometry of the fractures departs strongly from the idealized radially growing penny-shaped fractures. Instead, we observe fracture extension initially vertically downward developing later along an E-W oriented plane. The flow geometry is rather one-dimensional and pipe-like, likely formed by natural fracture intersections (Evans et al., 2005). In addition, during experiment HF2, multiple fractures at
the early time of the injection were observed. Evidence for these early fractures not far apart from each other are given by two tensional signals in strain monitoring borehole FBS1 at a depth of 31.8 and 33.0 m (Fig. 10b).

Generally, this confirms the strong influence of pre-existing fractures and fracture intersections on the structure of the hydraulic fracture propagation. It is furthermore noteworthy that the approximation of a penny-shaped fracture may not be appropriate in our case as the seismic clouds indicate a unidirectional instead of radial fracture growth (compare Figure 8). Therefore,
multiple fractures should be considered to advect the fluid rather than a single fracture. Considering the fracture geometry with two different orientations (HF2, HF3) a leak off point may be estimated, where the fluid starts to follow the pre-existing fracture network striking in E-W direction. This leak off point seems to be at a distance of 10 m from the injection point (Fig. 15). Further away from the injection point no seismic activity was observed. With ongoing hydraulic fracturing, more and more pre-existing fractures will be connected, which results in an increase of swept volume and a decrease fracture fluid pressure.
This decreases the efficiency to create new fractures and therefore seismicity decreases until enough overpressure is created to shear along pre-existing fractures or continue to grow tensile fractures. The interpretation agrees with the conceptual model for the HF experiment described as primary fracturing with shear stimulation leak off by McClure and Horne (2014).

The maximal number of localized seismic events from the HF experiment is 730 and the one from the minifracs is 646. However, seismic activity seems to be larger for the MF given the comparably smaller injection volume of 10 l compared to
>1000 l for the HFs. Only experiments HF2 and HF8 reach a number above 100 localized seismic events (Fig. 13). A possible reason may be stress relaxation from the HS experiments (Krietsch et al., 2019a) executed 5 months before the HF experiments. This would mean that shear relieved in the previously stimulated rock volume prevents new seismicity to occur. However, it seems that the HF experiments have also been conducted in a stress field that is naturally different from the MF locations.





## 6.4 Permeability creation by HF

All the performed hydraulic fractures generated a transmissivity increase of about three orders of magnitude. This increase remains after the pressure is relieved and in this sense is permanent, i.e. is not related to the transient fracture opening under high-pressure injections. This indicates that the fractures never completely close back when the pressure is relieved. The final

transmissivity matches one of an unstimulated fracture in our rock mass (Jalali et al., 2018a). The final transmissivity correlates positively with the final injectivity, which characterizes the newly created hydraulic fracture. Considering the injectivity value from the pressure-controlled step test has the advantage that it is directly related to the bulk mass compliance (rock mass compliance and fracture compliance) at short timescale. In addition, we can characterize the lift-off (herein: jacking pressure) of the intersecting fracture and the injectivity after jacking in the injection interval. The drawback is that the flow is only

at a quasi-steady state. For transmissivity values, we used constant head injection and pulse injection tests at low pressure ($< 0.6$ MPa). This method is time-consuming and it does not account for mechanical effects but has the advantage to reach steady-state conditions, which allow a characterization of the bulk volume farther away than the pressure-controlled step test.

Hence, both methods are mandatory to understand the change in the flow field and the permeability. Considering an EGS system with an injection and a production interval, then a certain injection pressure and probably a back pressure on the

production interval is applied. Both injection and production intervals are able to change the hydro-mechanical parameters of the nearby fractures. Then the injectivity value is able to describe this interaction, where the transmissivity describes the overall bulk volume of the connected fracture network. In our experiment, all the newly created hydraulic fractures (except HF6) were mechanically closed after the reservoir was depressurized. Low-pressure constant head injection was insufficient to open the closed hydro-fracture. Thus, the new created transmissivity is only linked to the new hydraulic fracture intersecting

the wellbore. In term of scaling we estimate transmissivity only on single wellbore and no cross-hole tests were performed to estimate the new fracture network transmissivity. Therefore, the presented values of the final transmissivity are rather small.

## 6.5 Stress heterogeneity

For HF2 and HF3, the initial plane fitted through seismicity cloud (compare Fig. 14c) dips to NE and E, respectively, but rotated at a later time to inclined South dipping orientation. One possible interpretation of this rotation is that the growth is initially

controlled by the perturbed stress field as described in (Krietsch et al., 2019b), i.e. with the minimum principal stress dipping to the WSW, while the later growth is controlled by the unperturbed stress field with the minimum principal stress dipping N. The result show that the minimum and intermediate stress magnitude are similar and allow to switch position.

For experiments executed south or north of S3 the ISIP stabilizes around 5 MPa and only deviates slightly. The jacking pressure is very similar in magnitude compared to the ISIP for the experiments south of S3 (see Fig. 5). The situation differs

for the experiment north of S3 as the magnitude of the jacking pressure is smaller. It reaches a value between 60-80% of ISIP. The similarity of jacking pressure and ISIP is a strong indication that they reflect the minimum principal stress magnitude, but further analysis is necessary to rely on that, which is not part of this publication.





Approaching the shear zones S1 and S3 changes the stress observations significantly. For the HF experiments, it is best described by the perturbed stress field for small volumes of injection, i.e. frac and first refrac cycle. Fig. 16a shows the perturbed stress state as Mohr circles for the newly created hydraulic fractures from the ATV log (triangles) and the best fit plane from seismicity (squares). Assuming, higher perturbation during the breakdown cycle creating the new hydraulic fracture intersecting

the borehole agrees well with the higher overpressure. Propagating the hydraulic fracture further away results in a decrease in the overpressure in the perturbed stress field (Fig. 16b) but an increase in the unperturbed one (Fig. 16c). Therefore, the unperturbed stress state cannot describe the behavior of the hydraulic fracture at the early time. The orientation of the seismic cloud is sub horizontal towards South considering all located seismic events (i.e. RF2). This reorientation of the fracture is controlled by the leak off into the pre-existing fracture network striking in E-W direction. Note, that the density of pre-existing

fractures is increased approaching the S1 and S3 shear zones. At this point, the best estimate of the stress state can be described by the unperturbed stress field assuming smaller stress magnitudes. Fig. 16d shows the situation in terms of minor and major principal stress axis being reoriented approaching the shear zones. The perturbed stress field is a consequence of the pre-existing fractures related to S1 and S3 and scatter the perturbed one. We assume this re-orientation of the fracture depends directly on the connectivity of the pre-existing fracture network.

## 15   7   Conclusions

In this paper, the spatial and temporal evolution of rock deformation, transient fracture fluid pressure, and seismicity during six intermediate scale hydraulic fracturing experiments are presented. One of the key findings of this work is that the fracturing processes are strongly influenced by site specific characteristics, natural fractures and local heterogeneities. This is to be expected in any rock masses since heterogeneities are always present in natural media. If the details of the observations

presented in this paper are site specific, the overall processes and behaviour are likely reproducible at any site and we attempt to formulate these general behaviours in the following conclusions:

1. The creation of new fractures at the borehole is clearly visible on borehole wall acoustic images. The orientation of the fracture traces are however variable and do not provide a good estimation of the independently measured far field stress orientation. The variability of the fracture trace could be explained by rock strength anisotropy (Dutler et al., 2018). It

could also be influenced by packer stresses or local stress heterogeneities induced by natural fractures as we observed that the new fracture trace extend below the packer and often abut against natural fractures. Our data set doesn't allow us to determine if the fractures initiate below the packers or in the injection interval, although since the packer pressure is always maintained above the interval pressure to insure sealing, the former is not unlikely.

2. The growth of the hydraulic fractures is strongly influenced by natural fractures. This leads in the details to complex

geometry departing from theoretical mode I fracture geometries. Our data highlights the simultaneous growth of parallel fracture strands. It also suggest channelized growth (pipe-like geometry) instead of planar growth.





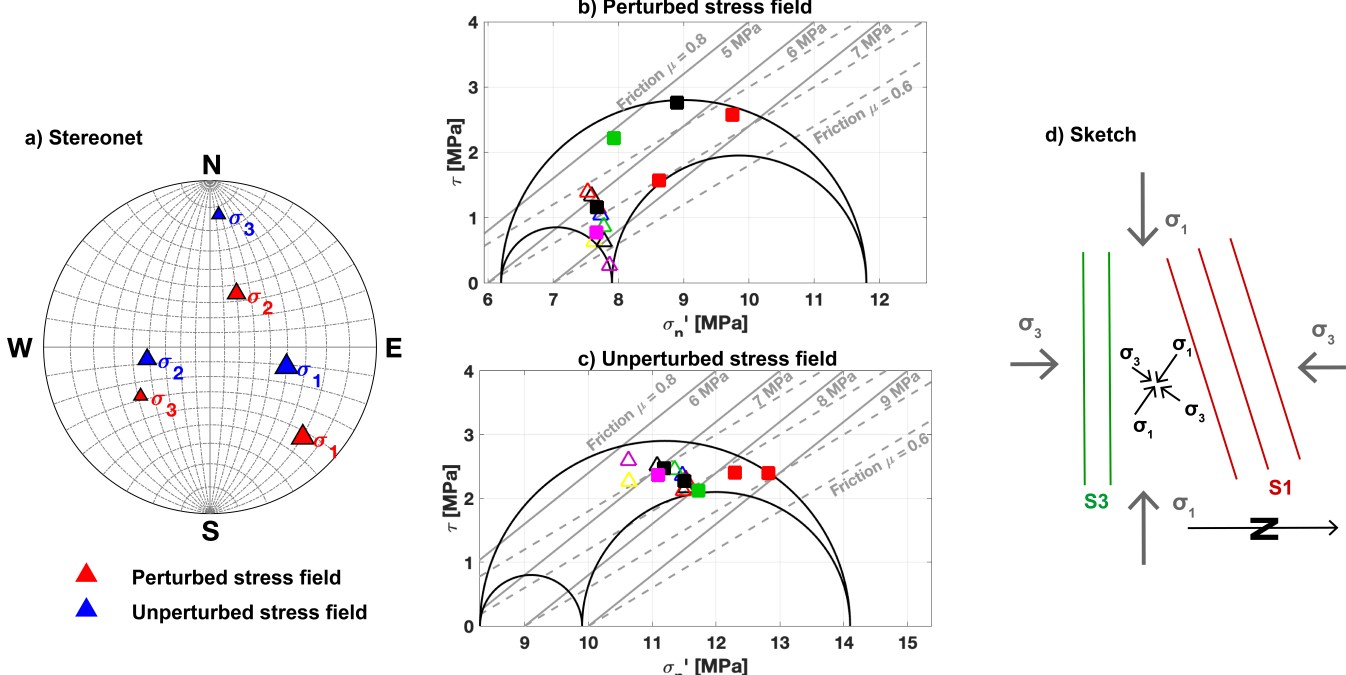

**Figure 16.** a) Stereonet with perturbed and unperturbed stress field and Mohr-Coulomb diagram representing the perturbed b) and unperturbed c) stress field (including hydrostatic pressure of 0.3 MPa) as Mohr circles. The failure limits assuming a friction coefficient of 0.6-0.8. Overpressure of 5 to 9 MPa are shown. The new created hydraulic fractures from the ATV log (triangle) and the best fit plane from seismic are presented with computed normal stress and shear stress. d) The sketch indicates the change from unperturbed (grey arrows on the outside) to the perturbed stress field (black arrows inside) approaching the two different shear zones S1 and S3.

3. The deformation field associated with fracture growth is also complex showing local extension on the fracture path that is compensated by compression in the surrounding rock mass. Such deformation pattern is strongly coupled to the pressure field variations associated with the injections.

4. Despite the complexity of the local fracturing processes, the spatial distribution of microseismic events associated with
   the fracture growth seems to be predominantly controlled by the stress state. In that regard, we observe rotations related
   to stress perturbations that have been highlighted within our experimental volume (Krietsch et al., 2019b). A comparison
   of deformation and pressure monitoring within the rock mass and the location of microseismic events suggest that
   the activation of seismic events lags slightly behind the propagation of the pressure front. This suggest that treatment
   size estimated by microseismic mapping could underestimate the actual rock volume affected by hydraulic fracturing
   processes.

5. The number of seismic events associated with each fracture treatment varies a lot. It is not related to the injected volume,
   but seems to be influenced by heterogeneities in stress and flow with our experimental volume.





6. The heterogeneities lead also to distinct behaviour in terms of pressure responses and flow: tests performed North or South of the brittle-ductile shear zone S3 respond differently. We associate these differences principally with two effects. Firstly, the shear zones have impacted the stress state locally and thus in turn this affect the fracture propagation. Secondly, the shear zones and the associated fracturing influence the flow in the experimental volume. When the hydraulic fracture grows, it connects and leaks into the pre-existing fracture network and thus at some point the energy required to create new fracture surfaces isn't sufficient and tensile fracture stops. The flow is then largely dominated by the natural fracturing. In that regard, the brittle-ductile shear-zone S3 acts as main drain and constant pressure boundary while connection to the S1 shear-zone and associated fracturing provides larger storage possibilities.

7. A significant increase in transmissivity of 2-4 order of magnitudes from well test before and after HF was observed. The final transmissivity correlates positively with the final injectivity obtained from the pressure-controlled step tests. Such permanent permeability increase is not consistent with pure mode I fractures in a homogenous media that would close back after depressurization. The heterogeneities and associated complexity of the hydraulic fractures probably favors the permanent transmissivity gain. The final transmissivities are comparable with the unstimulated natural fractures in our rock mass (Jalali et al., 2018b) which support the conclusion that the connectivity to the natural fracture system controls the final transmissivities.

*Data availability.* The Grimsel ISC Experiment Description is available at https://doi.org/10.3929/ethz-b-000310581 and the hydromechanical data set from the Grimsel ISC hydraulic fracturing experiment is available at https://doi.org/10.3929/ethz-b-000328270. The data from the minifrac experiments are available at https://doi.org/10.3929/ethz-b-000217536.

*Competing interests.* The authors declare that they have no conflict of interest.

*Acknowledgements.* The ISC is a project of the Deep Underground Laboratory at ETH Zurich, established by the Swiss Competence Center for Energy Research - Supply of Electricity (SCCER-SoE) with the support of Innosuisse. Funding for the ISC project was provided by the ETH Foundation with grants from Shell and EWZ and by the Swiss Federal Office of Energy through a P&D grant. Hannes Krietsch is supported by SNF grant 200021_169178. Linus Villiger is supported by grant ETH-35 16-1 and Nathan Dutler is supported by SNF grant 200021_165677. We thank Gerd Klee and his staff from MeSy Solexperts, Bochum (Germany) for their good collaboration and helpful discussions. The Grimsel Test Site is operated by Nagra, the National Cooperative for the Disposal of Radioactive Waste. We are indebted to Nagra for hosting the ISC experiment in their GTS facility and to the Nagra technical staff for onsite support.



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
