# Peer review of "Hydraulic fracture propagation in a heterogenous stress field in crystalline rock mass"

_Solid Earth, 2019_

## Referee Comment (RC1) · Anonymous Referee #1 · 16 Aug 2019

I do not have much to say about this paper which describes a well designed and interesting experiment. I have a few minor comments: 1. the ms is well written, but very long. Is there any way to shorten it? 2. the colors in Fig.8 are not adequate! Many dots are hardly visible. 3. One may criticize the discussion section by noting that there is very little quantitative exploitation of the data. For instance, it would be nice to relate the transmissivity increase to the created fractures in a quantitative way.

---

## Referee Comment (RC2) · David McNamara (Referee) · 21 Aug 2019

This is a very interesting and really useful study to the geothermal community and the rock-deformation community given the rarity of such experiments in-situ and good monitoring of their impacts. This is highly novel work in that respect, and should be of great use and interest to both academia and industry. I strongly encourage publication of this manuscript after some minor revisions.

Most of my critiques are minor and I am sure can be easily addressed by the authors. I do have one pressing concern with the Discussion, particularly with how it is structured and written. Given the value of such studies like these, and their scarcity, I think the paper would really benefit from a rewrite of the discussion to more clearly lay out

the authors arguments and how they are supported by the data they have acquired. Currently the discussion is at times difficult to follow and a slight rewrite/reorganisation, and some careful rephrasing of parts would immensely improve this manuscript. I have tried to provide more detail below on where I think some useful improvements could be made, to assist the paper outputs in making a bigger impact.

There are places in the discussion where a strong conclusive statement or argument is made by the authors for a particular theory they support. While this is great to see, I would be interested to see the authors take some space in the paper to discuss possible alternative interpretations to their preferred theory based on the observations and present the arguments for and against them.

I have made several comments below specifically about stress features in borehole images that I would like the authors to address.

The results presentation in the manuscript is quite lengthy mostly due to extended and suffers textual description of the results. Perhaps the authors could look at refining and reducing the presentation/description of the data in the manuscript text, referring more heavily to their excellently designed figures? While this would not be essential for the eventual publication of this paper it would improve its readability.

The figures are fantastic though I have only suggested minor edits to improve their usefulness.

Minor correction suggestions:

Page 2, Line 6 – References to support the statement of massive HF technology in petroleum and geothermal should be provided here.

Page 2, Line 6 – 'Deep' geothermal is a little subjective, some conventional geothermal that requires no HF goes to depths of up to 4km. I would suggest refraining from the use of deep in this context and sticking with enhanced geothermal systems which are directly involved with HF processes.

Page 2, Line 12 – Another example of confluence of deep geothermal with enhanced geothermal. I would advise avoiding this as it does not fit with all examples of 'deep' geothermal globally, nor with all EGS which can also be explored at more shallow depths.

Page 5 Line 19 – what is the dip direction of the foliation?

Page 6, Figure 1 – While northing and easting are noted on the model axes, a N-direction arrow would make these much easier for a reader to visualise the 3D structural geometry. The blue circles in d) are very difficult to see.

Page 6 – What are the two fracture systems mentioned for the d-b S3 shear zone? A reference to Figure 2a would be useful here.

Page 7, Table 1 – The header suggests that the orientations are displayed as dip and dip direction, yet are written in the table the other way around. Make the table header consistent with the way the information is displayed.

Page 7, Line 7 – When you state the perturbed stress field are you referring to the stress state within the fractured zone associated with the S3 shear zone, or does it extend beyond this? It becomes clearer to the reader later in the paper but would be better to clarify this at this point.

Page 8, Figure 2 – a) what are the coloured triangles in stereonets. It would be useful to have the symbols used here be the same as those used in 2c to make the link between these two diagrams more obvious. B) Is the stress state in the corner the arrangement of the perturbed region or the unperturbed? A north arrow on the 3D model (the left diagram) would make it easier to read than having to go off the axes. C) Black lines with pore pressures on them are not explained in the caption.

Page 9, line 3 – By 'intact injection intervals' do the authors mean intact lithology with no brittle deformation visible? This could be written to be clearer to the reader.

Page 9, line 8 – This statement is only true for some of the fracture density peaks, or

some measurements of Tsz but not all. The correlation of the two does not always match up, perhaps the authors could rethink the strength of this statement and explain the variability observed here in the correlation, or refer to a paper that has discussed this?

Page 15, Figure 4 – Be specific in the caption about what the black lines on graph 4a represent.

Page 16, Figure 5 – The labelling of which bar chart is backflow and which is injection could stand to be presented a little more clearly. It was difficult to read at times.

Page 17, Line 1 – repetition of 'the'.

Page 18, Line 22 – change 'posterior' to 'post'.

Page 18, Figure 6 – The image log for HF1 (post-test) shows a pre-existing fracture in the test interval, albeit towards the bottom of the HF interval that was not imaged in the initial televiewer log. This needs to be addressed by the authors for any potential effect on the HF test results, or explained as an induced structure with proof that it is one. Are there other mechanical weakness in these intervals beyond fracturing (layering etc.), some of the image logs (HF5 in particular) suggest there may be a fabric in these depth intervals, is there any information from core about layering etc.? If so, has the possible effect of these mechanical weaknesses in the 'intact' rock on HF testing been considered? Do the orientations of the induced fractures agree with the existing stress field orientations i.e. strike perpendicular to sigma 3 (depending if they are DITF or petal centreline fractures)? It would be good to see this information presented somewhere in the manuscript or supplement for comparison to already published stress data. The authors mention later that some of the induced fractures formed during testing are opened along a foliation orientation – this is very important as the sigma 3 value derived from these tests is that required to open a plane of weakness which may not necessarily represent the tectonic stress magnitude. I encourage the authors to include further discussion of this and the potential implications of it on their

findings. Currently I think not enough consideration has been given to this.

Page 19, Line 4 – en echelon morphologies for the induced fractures are mentioned. These are usually a result of a well deviated with respect to the orthogonal stress tensor. Have corrections for the stress orientations determined from these image logs been run to account for the effect of the deviated well? What is the deviation of the wells in the HF intervals?

Page 19, Line 7 – Do the authors mean 730 events were located in D space? Change the word localised to located if so.

Page 19, Line 12 – Villiger et al. reference incomplete, missing the year.

Page 19, Figure 7 – Does the colour intensity of the seismicity dots reflect anything or is this simply due to a number of points of the same colour overlapping on the diagram? What are the 3D cylinders (various colours) around the wells?

Page 20, Line 30 – How was the best fit plane determined? The fit looks weak on the figure, how good is it as a representation? Can the authors comment on this?

Page 22, Figure 9 – the coloured points on the time series and the polar plots do not match (one is outlined the other is not). A small consideration but it would be nice to make these consistent between diagrams.

Page 20 – 'Injection executed south of S3 show in general smaller magnitudes than the one' – can the authors clarify what is smaller in magnitude? Is it tilt or seismicity?

Page 21, Line 6 – 'tiling of the tunnel floor away' – clarify away from what. This sentence is very difficult to follow. Please consider rewording for clarity.

Page 28, Line 15 – Villiger et al reference missing a year.

Page 29, Line 20 – have a similar trace orientation 'to' the HF experiments.

Page 30, Figure 14 – the labelling of the points on the stereonets is incomplete in

places and confusing. A key in the corner of this figure would be more useful perhaps? Please revise.

Page 32, Line 5 – is as 'follows'.

Page 32, Line 6-9 – An important part of this section of the manuscript. I advise the authors to rewrite this sentence or two with an emphasis on clarity of meaning, as it does not currently read as well as it could.

Page 32, Line 11-18 – This section could benefit from a rewrite with the aim of ordering the arguments made in a clearer and easier to read fashion. It is currently confusing and is not quite making the point the authors are trying to make which is that these results support the theory of the S3 being hydraulically connected to an open system with respect to fluid flow.

Section 6.1 – It is my opinion that this section suffers a little due to its structure. I think if the authors took some time to order their arguments better the scientific findings of this section would be more apparent to a reader. As it currently reads, it is confusing and I think it is very important the authors consider how to rephrase this section to better highlight their arguments.

Page 32, Line 33 – Does this actually cast doubt on stress orientation from well data only or is the 30-degree rotation result from another process such as pre-existing weaknesses affecting the propagation of the induced fractures? I would like to see a discussion on the other arguments for this observed orientation difference between the seismic cloud and the stress orientation from borehole data, with arguments for and against each before coming to a final conclusive statement such as the one currently made.

Page 33, Line 1 – How is the trace of a fracture scattered? I encourage the authors to revise the clarity with which they have written sections and statements of their discussion as they are often difficult to follow, and difficult to relate to any conclusion or

argument being made.

Page 33, Line 5-8 – I am unsure why the authors state the trace of the induced fractures are complex, they are actually quite clear in form and morphology from the images provided in figures. What is more complex is where these fractures nucleated during HF tests, i.e. at the abutting natural fractures or within the sealed section of well itself? The interaction of the induced fracture with natural fractures outside the packer would seem to infer that the fracture propagated vertically until it met a pre-existing structure and then ceased to propagate vertically. However, that is difficult to prove without directly observing the induced fracture growth. Finally, the comment on stress orientations from induced features being misleading in deviated wells is old news and there are numerous studies (Hickman, McNamara, Davatzes, Barton, Zoback) that have addressed this issue in a number of geothermal well scenarios. For example, McNamara et al., 2015 (Rotokawa geothermal borehole imaging paper) calculated minimal effect on using induced features for stress orientation in geothermal wells deviated at angles up to ∼22 degrees from vertical. I would like to see the authors acknowledge some of the points raised here in this section of the discussion as they have important implications to their work.

Page 33, Line 11 – to some 'extent'.

Page 33, Line 11-13 – Have the authors considered that the induced fractures do not grow in a penny-shape fashion due to the influence of the curved free-surface of the borehole wall?

Page 39, Line 22-34 – It is standard practise to assume that induced features in borehole images over small well lengths are not representative of the regional far field stress orientations but are in fact likely representative of localised effects or heterogeneities of the stress field orientation. This is reflected in the quality ranking system developed for such data by the World Stress Map project. So, while the data presented here by the authors may not agree or align with expected regional far field orientations they are

likely still providing accurate information on the local perturbations in the orientation.

Page 39, Line 29 – Please check the English of this sentence.

Page 40, Figure 16 – Please include a key for the coloured symbols on the Mohr space diagrams.

Page 40, Line4 – Conclusion 4 and conclusion 2 sound contradictory. 2: The growth of the hydraulic fractures is strongly influenced by natural fractures. 4: the spatial distribution of microseismic events associated with the fracture growth seems to be predominantly controlled by the stress state. While I appreciate these are subtly different data being measured I would encourage the authors to comment somewhere on which aspect, if any, is more controlling on fracture growth (stress or pre-existing weaknesses), or discuss somewhere the link between the stress, natural fracturing, and the effects this can have on induced fracture propagation.

---

## Author Comment (AC1) · 4 Oct 2019

**Response to the reviewers**

We would like to thank the reviewers for their valuable time and useful contribution. We appreciate the inputs they have given and their critical assessment of our work. Their suggestions have improved the quality of the article. In the following we address their concerns point by point.

[Figure]

**Reviewer 1**

**Reviewer Point P 1.1** — The results presentation in the manuscript is quite lengthy mostly due to extended and suffers textual description of the results. Perhaps the authors could look at refining and reducing the presentation/description of the data in the manuscript text, referring more heavily to their excellently designed figures? While this would not be essential for the eventual publication of this paper it would improve its readability.

**Reply**: We reduced the extended results presentation inclusive textual description to improve the readability of the manuscript as followed:

**We deleted:**

- **Page 22, line 25-27**: This zone is located West of the injection point. During the experiment, tiltmeter T1 dips away from shear zone S1. Tiltmeter T2 tilts away from the zone S3 during the first refrac cycle and reorients itself with further injection (RF2) indicating a transient shift along the S3 shear zone.

- **Page 22, line 31-36**: The tiltmeter signals are interpreted as secondary deformation fields, where T2 indicates transient movement of the S1.2 due to dextral shearing along S3. Tiltmeter T1 (located south of S3.2) has a delayed response and indicates movement towards the shear zone S3.1 during the second refrac RF2 cycle. Then, it starts to reorient itself during shut-in and bleed-off time indicating mass diffusion along the preferential flow paths (fracture associated with zone S1.0). Reversible versus irreversible deformation for the tilt meter data will not be discussed in here due to long term changes observed in the time-series (i.e. tides, seasons).

- **Page 25, line 4-9**: show tension during the first refrac cycle in borehole FBS1 at a
depth of 31.8 and 33.0 m and compression at a depth of 42.2 m. At the same time when the sensor at 42.2 m starts to release compression, the sensor in FBS2 at 43.3 m starts to show tension and the two other sensors in FBS1 show a decrease in tension. During shut-in, all selected FBG sensors show an immediate decrease in tension except the sensor at 42.2 m in FBS1, which still shows an increase in tension until the injection interval was bled off to release the pressure. The strain sensors along the borehole FBS1 show the largest magnitude compared to the sensors in the other boreholes.

- **Page 25, line 15-19**: The FBG sensors in FBS3 show no signal above 30 m. A small compressional signal is observed between 30 and 35 m and tensional signals below 30 m. The tensional signal is observed in the S3 shear zone, indicating fluid flow towards south. The brittle-ductile shear zone S3 response differs approaching the deeper volume, where FBS3 intersects the S3 shear zone compared to FBS1 (no response).

- **Page 25, line 26-30**: During the first refrac cycle, the two aforementioned sensors show small compression before they reverse to tension. This can be interpreted as propagating fluid front that reaches the brittle-ductile shear zone S3, which drains the injection fluid towards the AU-tunnel. The behavior of sensor at 24.0 m in FBS2 is similar but delayed. It shows compression at the beginning which changes to compression at the end of refrac cycle RF1. At the same time, the sensor at 20.0 m in FBS2 shows a sharp drop and starts to compress.

**Reviewer Point P 1.2** — Page 2, Line 6: References to support the statement of massive HF technology in petroleum and geothermal should be provided here.

**Reply**: We added two well known textbook references to support the statement of massive HF technology for both, petroleum and geothermal applications.

The following changes are made:
**Page 2, Line 5:** Massive hydraulic fracturing technology is often used in the oil and gas industry (Economides and Nolte, 2000) and in applications in the context of enhanced geothermal projects (Brown et al., 2012).

**Reviewer Point P 1.3** — Page 2, Line 6: 'Deep' geothermal is a little subjective, some conventional geothermal that requires no HF goes to depths of up to 4km. I would suggest refraining from the use of deep in this context and sticking with enhanced geothermal systems which are directly involved with HF processes.

**Reply**: We agree with the reviewer, that the terminology 'Deep geothermal projects' is confusing and actually refers to any type of deep geothermal energy plant. Therefore, we will use the term 'enhanced geothermal system' for geological settings with natural insufficient permeability, such that hydraulic stimulation is used to promote flow between different boreholes.
The following change is made:
**Page 2, Line 6:** ... context of enhanced geothermal projects.

**Reviewer Point P 1.4** — Page 2, Line 13: Another example of confluence of deep geothermal with enhanced geothermal. I would advise avoiding this as it does not fit with all examples of 'deep' geothermal globally, nor with all EGS which can also be explored at more shallow depths.

**Reply**: According to comment P1.2 and this comment, we removed the term 'deep'.
The following change is made:
**Page 2, Line 12:** The context of our study is the exploitation of deep geothermal energy...

**Reviewer Point P 1.5** — Page 5 Line 19: What is the dip direction of the foliation?

**Reply**: To increase the readability of the manuscript we added in brackets how to read dip direction and dip, at the first place it occured.
The following change is made:
**Page 5, Line 20:** ... resulting in a pervasive foliation oriented 157/75° (dip direction/dip given in xxx/xx°)

**Reviewer Point P 1.6** — Page 6, Figure 1: While northing and easting are noted on the model axes, a N- direction arrow would make these much easier for a reader to visualise the 3D structural geometry. The blue circles in d) are very difficult to see.

**Reply**: We changed Figure 1 accordingly to the suggestions:

- We added to all subfigures a N-direction arrow, next to the AU tunnel for easier visualisation.

- We enhanced the radius of the FBG circles in subfigure d) for better visibility.

- We skipped the S3 brittle-ductile shear zone from subfigure b) for easier readability and added a grid to the plot.

The following change is made in the caption of Figure 1:
b) The AU cavern, the AU tunnel, the S3 shear zone and the injection interval locations in the two injection boreholes are shown and numbered.

**Reviewer Point P 1.7** — Page 6: What are the two fracture systems mentioned for the d-b S3 shear zone? A reference to Figure 2a would be useful here.

**Reply**: We added the Fig. 2a reference to the mentioned two fracture systems. At this point we will refrain from a further description, as this will follow later in section 2.2.2 Hydraulic characterization.

The following change is made:
**Page 7, Line 2:** ...different fracture systems (Fig. 2a).

**Reviewer Point P 1.8** — Page 7, Table 1: The header suggests that the orientations are displayed as dip and dip direction, yet are written in the table the other way around. Make the table header consistent with the way the information is displayed.

**Reply:** We fixed this error and made the table header consistent. In addition, we precised the description of the perturbed stress rotation.
**Page 7, Line 14:** ...1) a primary 30° clockwise rotation accompanied with a decrease of the major principal stress dip and 2) a permutation of the intermediate and minimum stress axis.

**Reviewer Point P 1.9** — Page 7, Line 7: When you state the perturbed stress field are you referring to the stress state within the fractured zone associated with the S3 shear zone, or does it extend beyond this? It becomes clearer to the reader later in the paper but would be better to clarify this at this point.

**Reply:** Prior to these HF experiments, we did not know how far the perturbed stress field can be extended. During the stress characterization campaign, we did only test from the tunnel towards the S3.1 shear zone but not beyond. The new results presented in our paper shows that the perturbed stress state can be extrapolated beyond the S3 brittle-ductile shear zones towards the S1 shear zones.
The following is added to the manuscript:
**Page 7, Line 13:** This study will show that the perturbed stress state can be extended from the S3 brittle-ductile shear zone towards the S1 shear zone.

**Reviewer Point P 1.10** — Page 8, Figure 2: a) what are the coloured triangles in stereonets. It would be useful to have the symbols used here be the same as those

used in 2c to make the link between these two diagrams more obvious. B) Is the stress state in the corner the arrangement of the perturbed region or the unperturbed? A north arrow on the 3D model (the left diagram) would make it easier to read than having to go off the axes. C) Black lines with pore pressures on them are not explained in the caption.

**Reply**: a) The three stereonets show fractures (black points) depending on the identified structures which are classified in protolith, S1 shear zone and S3 shear zone. A fracture was identified accordingly to the borehole depth and allocated. The colored triangles in stereonets indicate the pole point of the interpolated shear zones, which are indicate now. These triangles are actually linked to Figure 2b. Figure 2a and 2c are linked via the legend of Figure 2c, which is the same than the titles in Figure 2a. Figure 2c indicate fractures from the three different classified types using different colors and symbols for better readability. We changed the caption of the figure accordingly.
b) We added a N-direction arrow in the 3D model (left figure) and indicate the unperturbed stress state in block model (right figure).
c) We added this to the caption.

The following change is made in the caption of Figure 2:
The identified and allocated fractures from borehole logging in all 15 boreholes are presented to indicate possible failure at a specific pore pressure (grey solid lines).

**Reviewer Point P 1.11** — Page 9, line 3: By 'intact injection intervals' do the authors mean intact lithology with no brittle deformation visible? This could be written to be clearer to the reader.

**Reply**: We included the term intact lithology to the definition and changed the term discontinuity by brittle deformation.

The manuscript is adapted as followed:
**Page 9, Line 7:** intact injection intervals (defined here as intact lithology by the absence of visually detected brittle deformation on cores and borehole image logs)

**Reviewer Point P 1.12** — Page 9, line 8: This statement is only true for some of the fracture density peaks, or some measurements of $T_{SZ}$ but not all. The correlation of the two does not always match up, perhaps the authors could rethink the strength of this statement and explain the variability observed here in the correlation, or refer to a paper that has discussed this?

**Reply**: Following the reviewer advice, we tempered our statement. Going to the details of the hydraulic heterogeneities of the tested rock mass is beyond scope in this context. We refer to Brixel et al. (2019) for more details.

The manuscript is adapted as followed:
**Page 9, Line 12:** The estimated transmissivities are in good correlation with the fracture intensity in the injection borehole INJ1 (Figure 3d, left), although the spatial resolution of our hydraulic measurements do not permit to capture all fracture intensity peaks. These measurments are also dependent on the natural heterogeneities, particularly related to the S1 shear zones. More details on these complex relations can be found in Brixel et al. (2019).

**Reviewer Point P 1.13** — Page 15, Figure 4: Be specific in the caption about what the black lines on graph 4a represent.

**Reply**: The black solid lines indicate the limiting pressure for tests executed north (HF1/HF2) or south (HF3) of the S3 brittle-ductile shear zone. There was an error in the figure indicating the limiting pressure of test HF8 (XSW) instead of HF3 (W). The different fluids have different dissipation mechanism and therefore it is more convenient

to compare tests with the same injection fluid (in our case: water). The limiting pressure occurs above 30 l/min for test HF1 and HF2. For test HF3 we see a limiting pressure above 20 l/min for test HF3, which goes up to 60 l/min. Then it starts to increase again due to either short times between flow step or increasing viscous dissipation. Test HF8 is interpreted to show as well a limiting pressure above 50 l/min. Due to the XSW injection, which increases the viscous dissipation the limiting pressure is bigger than for HF3 and reaches 7.8 MPa.

The caption of Figure 4 reads now:
a) Presents the injection flowrate $q_{inj}$ vs interval pressure $p_{inj}$ at pseudo steady-state from the second refrac cycle RF2 and pressure-controlled step test SR (similar to Fig. S1 in the supplementary information) for all HF experiments. The black solid lines indicate a limiting pressure behaviour for HF1/HF2 and HF3. (b-d) Highlights the pressure drop when changing from XSW to water with similar flow rate.
**Page 16, Line 7:** (HF3 and HF8) and is interpreted...

**Reviewer Point P 1.14** — Page 16, Figure 5: The labelling of which bar chart is backflow and which is injection could stand to be presented a little more clearly. It was difficult to read at times.

**Reply**: We added a labelling to each bar for an intuitive reading.

**Reviewer Point P 1.15** — Page 17, Line 1: repetition of 'the'.

**Reply**: Fixed.

**Reviewer Point P 1.16** — Page 17, Line 22: change 'posterior' to 'post'.

**Reply**: Fixed.

**Reviewer Point P 1.17** —

Page 18, Figure 6: The image log for HF1 (post-test) shows a pre-existing fracture in the test interval, albeit towards the bottom of the HF interval that was not imaged in the initial televiewer log. This needs to be addressed by the authors for any potential effect on the HF test results, or explained as an induced structure with proof that it is one. Are there other mechanical weakness in these intervals beyond fracturing (layering etc.), some of the image logs (HF5 in particular) suggest there may be a fabric in these depth intervals, is there any information from core about layering etc.? If so, has the possible effect of these mechanical weaknesses in the 'intact' rock on HF testing been considered? Do the orientations of the induced fractures agree with the existing stress field orientations i.e. strike perpendicular to $\sigma_3$ (depending if they are DITF or petal centreline fractures)? It would be good to see this information presented somewhere in the manuscript or supplement for comparison to already published stress data. The authors mention later that some of the induced fractures formed during testing are opened along a foliation orientation – this is very important as the $\sigma_3$ value derived from these tests is that required to open a plane of weakness which may not necessarily represent the tectonic stress magnitude. I encourage the authors to include further discussion of this and the potential implications of it on their findings. Currently I think not enough consideration has been given to this.

**Reply**:

We thank the reviewer to point out the missed fracture in the initial televiewer log. We reanalysed the initial televiewer logs and revised Figure 6 in such that we changed to an older ATV log for the 'preHF1 amplitude log'. This was recorded after the HS experiment. Therefore, we can now address the fracture towards the bottom of the HF interval. The fracture is not observed in the preHF log. The orientation and placement

of the fracture indicate a packer induced fracture. This is also supported by the pulse
tests (prior to the HF experiment), which indicated no increased transmissivity for this
interval.

In term of mechanical weakness, we carefully analysed drill cores, optical and acoustic
televiewer to decide, where injection should take place ensuring no pre-existing frac-
ture was present in our test intervals. However, a pervasive feature for the Grimsel
Granodiorite present in our experimental volume is the presence of a clear overprint
from Alpine deformation. A foliation is often well developped (the rock could be named
Gneiss) and laboratory results indicate a transversal isotropic behaviour with direc-
tional dependency in Young's modulus, Poisson's ratio, tensile strength and fracture
toughness (see Dambly et al., 2019; Dutler et al., 2018). Thus the geometry of the cre-
ated fracture at the borehole wall will be influence by the stress state, the mechanical
transvers isotropy and the borehole orientation.

The stress state in our test volume is described in Krietsch et al. (2019a). Since the
magnitude of $\sigma_2$ and $\sigma_3$ are similar, it can lead to ambiguity on the hydraulic fracture
orientations. This, combined with the transverse isotropy of the intact rock mechanical
properties, leads to the observed variability of the observed fracture trace at the bore-
hole wall (see Fig. 14).

The manuscript is adapted as followed:

**Page 18, Line 20:** , which indicates new en-échelon structures for one side of the
fracture trace and mostly likely a packer induced fracture trace towards the bottom of
the HF interval.

**Reviewer Point P 1.18** — Page 19, Line 4: en echelon morphologies for the induced
fractures are mentioned. These are usually a result of a well deviated with respect to
the orthogonal stress tensor. Have corrections for the stress orientations determined
from these image logs been run to account for the effect of the deviated well? What is
the deviation of the wells in the HF intervals?

**Reply**: Our injection boreholes are oblique (309.57/33.52° (azimuth/dip°) for INJ1 and 332.28/43.65° for INJ2). They are very likely not aligned with the principal stress and that is why, as the reviewer rightly pointed out, we observe en-echelon fractures. We didn't compute theoretical en-echelon fracture angles for our boreholes. Note that to be meaning full, such computations should include the effect of transverse isotropy in mechanical parameters. In addition, stress heterogeneities will impact locally the exact angles and thus make such computations uncertain.

The manuscript is adapted as followed:
**Page 18, Line 21:** These structure is a result of the deviated borehole with respect to the perturbed stress tensor (Table 1).

**Reviewer Point P 1.19** — Page 19, Line 7: Do the authors mean 730 events were located in D space? Change the word localised to located if so.

**Reply**: Fixed.

**Reviewer Point P 1.20** — Page 19, Line 12 – Villiger et al. reference incomplete, missing the year.

**Reply**: Fixed and updated.

**Reviewer Point P 1.21** — Page 19, Figure 7: Does the colour intensity of the seismicity dots reflect anything or is this simply due to a number of points of the same colour overlapping on the diagram? What are the 3D cylinders (various colours) around the wells?

**Reply**: The colour intensity of the seismicity points does not change. The apperent change in intensity is due to the number of points overlapping on the diagram. The

open interval of injection are indicated by colored cylinders and labeled. For clarification, we added this to the caption of the figure.

The caption of Figure 7 is changed accordingly:
Seismicity clouds in side view a) and map view b) inclusive 3D cylinders presenting the open injection intervals in the two injection boreholes. Each experiment has the same color for seismicity points, the 3D cylinders and the labels.

**Reviewer Point P 1.22** — Page 20, Line 30: How was the best fit plane determined? The fit looks weak on the figure, how good is it as a representation? Can the authors comment on this?

**Reply**: In the supplementary information is a chapter with the title: 'S5: Fracture orientation determination from borehole logging and seismic plane analyses', which explains how the best plane was fitted and what the misfit is. We missed to link this to the chapter. Below, an excerpt of the supplementary information:
"For each HF experiment, we tried to fit a plane through the seismic cluster using an orthogonal distance regression criterion. The same approach was used for the minifracs by Gischig et al. (2018). The planes considering all located seismic events for test HF8 show a comparable orientation with the MF planes. The misfit of the plane for the test HF8 is 0.52 m. Test HF2 show two clusters with different oriented planes. The first cluster is located next to the injection borehole and occurs mainly through injection cycle RF1 and the second one is located further towards E during cycle RF2. One plane is oriented towards NNE with a dip of 83° and the misfit is 0.26 m and the other plane is oriented towards S with a dip of 76° (misfit: 0.34 m)."
The small misfits for the orthogonal distance regression indicate a good plane fit. The alternative approach looking on a stereonet does not indicate a good fit, as we compare here the seismic pattern from the injection point and looking outwards to the total stimulated volume. It actually shows that the hydraulic fracture is not made of a single

unidirectional fracture. The tendency of clustered points on the stereonet, which looks like linear structures are actually made of small planes, which are not aligned with the orientation of the averaged single fracture assumption.

The manuscript is adapted as followed:
**Page 22, Line 1:** In Figure 8 we present the best fit plane (for the method and results see S5 in the supplementary information) through the seismic cloud via girdle pattern...

**Reviewer Point P 1.23** — Page 22, Figure 9: the coloured points on the time series and the polar plots do not match (one is outlined the other is not). A small consideration but it would be nice to make these consistent between diagrams.

**Reply**: Fixed.

**Reviewer Point P 1.24** — Page 20: 'Injection executed south of S3 show in general smaller magnitudes than the one' – can the authors clarify what is smaller in magnitude? Is it tilt or seismicity?

**Reply**: We added this point and adapted the manuscript accordingly.

**Page 22, Line 20:** Injection executed south of S3 show in general smaller magnitudes in tilt than the one executed next to S1,...

**Reviewer Point P 1.25** — Page 23, Line 6: 'tilting of the tunnel floor away' – clarify away from what. This sentence is very difficult to follow. Please consider rewording for clarity.

**Reply**: We reworded this sentence.

**Page 22, Line 28:** Tiltmeter T1 of the experiments south of S3 (HF3 and HF8) indicates tilting of the tunnel floor away compared to the normal of S1.2 shear zone and T2 into the shear zone S3 indicating a small compressive component towards the zone S3.1. Tiltmeter T1 of the experiments south of S3 (HF3 and HF8) tilts away from the S1.2 shear zone and T2 tilts along the strike of the S3 shear zone.

**Reviewer Point P 1.26** — Page 28, Line 15: Villiger et al reference missing a year.

**Reply**: Fixed.

**Reviewer Point P 1.27** — Page 29, Line 20: have a similar trace orientation 'to' the HF experiments.

**Reply**: Fixed.

**Reviewer Point P 1.28** — Page 30, Figure 14: the labelling of the points on the stereonets is incomplete in places and confusing. A key in the corner of this figure would be more useful perhaps? Please revise.

**Reply**: In order to address the reviewer concern, we rearranged the positioning of the labels in the Figure. We see an advantage of keeping the labelling on the stereonets directly compared to a side legend, because in some situation, data overlapping can hide some of the information. This is the case for MF01 in subfigure d) which is hidden below data points MF03 and MF04. Having a label directly on the Figure allow to clearly hint to the presence of the MF01 data point. We clarify this also in the Figure caption. In addition to maintain consistency accross the paper and facilitate readability,

we consistently use the same color code for the experiments. That is why Figure 1b, 4, 7, 13, 15 and 16 have the same color code for HF (and MF if used).

The following change is made in the caption of Figure 14:
...Note that the MF01 data point is collocated with data points MF03 and MF04.

**Reviewer Point P 1.29** — Page 32, Line 5: is as 'follows'.

**Reply**: Fixed.

**Reviewer Point P 1.30** — Page 32, Line 6-9: An important part of this section of the manuscript. I advise the authors to rewrite this sentence or two with an emphasis on clarity of meaning, as it does not currently read as well as it could.

**Reply**: This comment is now addressed and the manuscript is amended accordingly (see the reply to comment P 1.32).

**Reviewer Point P 1.31** — Page 32, Line 11-18: This section could benefit from a rewrite with the aim of ordering the arguments made in a clearer and easier to read fashion. It is currently confusing and is not quite making the point the authors are trying to make which is that these results support the theory of the S3 being hydraulically connected to an open system with respect to fluid flow.

**Reply**: This comment is now addressed and the manuscript is amended accordingly (see the reply to comment P 1.32).

**Reviewer Point P 1.32** — Section 6.1: It is my opinion that this section suffers a little due to its structure. I think if the authors took some time to order their arguments

better the scientific findings of this section would be more apparent to a reader. As it currently reads, it is confusing and I think it is very important the authors consider how to rephrase this section to better highlight their arguments.

**Reply**: We revised Section 6.1 and changed the structure. We first discuss the hydraulic response, followed by the mechanical discussion. We also shorten the section and rephrased many of the sentences.

**Section 6.1 reads now:**
Hydraulically, two different behaviors were observed in experiments performed south of S3 (HF3 and HF8) respective north of the S3 (HF1, HF2 and HF6) shear zone. The differences consisted of lower pressure levels (fracture propagation, jacking and ISIP), larger recovered water volumes, and larger final injectivity for experiments north of S3. Our proposed explanation of this behaviour is as follows.
During the experiments performed south of S3, the new initialized HF was only able to propagate a short distance in intact rock before it connected to the densely fractured S3 shear zone, which is bounded by two metabasic dykes and therein two subnormal fracture systems are found. This high transmissive structure is able to drain the injected fluid from the HF towards the AU tunnel. None of the HF experiments performed south of S3 were able to fracture through this geological structure. This is due to two reasons 1) the structure large transmissivity drain the fluids and prevent further pressure propagation and 2) the stress variations in the structure vicinity are not favourable for further fracture propagation. This is further supported by the small fluid recovery (less than 2%) from the injection borehole for these injection tests, which suggests a strong hydraulic gradient forcing the injection fluid into the S3 shear zone and toward the AU tunnel.
The injections north of S3 (HF1 and HF2) were able to initialize and propagate an hydraulic fracture before leak-off dominated into the natural fractures associated with the ductile shear-zone S1. This less transmissive S1 shear zone can be described by a single fracture system, where one part of the fractures are capable to transport fluid

and another part only store the fluid (known as dead-end fractures). These tests are also more distant from the AU tunnel and thus in a volume isolated from its influence. Therefore, the observed fluid recovery from the injection location reaches values of 15.0-23.5% for the first two refrac cycles. The HF6 experiment is comparable with HF1 and HF2 except for the presence of a pre-existing fracture that influenced hydraulic fracturing initiation.

The tilt response reflect the hydraulic balance observations, i.e. the tilt magnitudes are smaller for the experiment south than north of S3. The FBG-strain sensors show both tensional and compressional signals. High tensional signals are related to fracture pressurisation and opening reflecting hydraulic connections. In response to fracture opening, some other segment of the rock mass need to compensate and this is captured by the observed compressional signals. An example of such behaviour is visible along FBS2 in Fig. 10f, with increasingly compressive signal from 5 to 16 m and from 40 to 26 m centered around and strongly tensional rock mass section at 20 m. The repartition of strain response is also influenced by the location of the injection south of north of S3. For the tests located south of S3, a tensional signal on the FBG sensors is predominantly observed in the S3 shear zone. The temporal evolution of strain signal also suggest modifications of flow path during the injections. Indeed, some sensors that initially react in compression shows later an abrupt reversal toward tensional strain, indicating that they are now part of the active flow system. In turn this influences previously active sections as sensor that were in tension stabilise or reverse, indicating a transfer of the flow portion towards the newly activated rock mass sections (Fig. 10).

For the injection locations north of S3, the geometry of our monitoring system implies that the distance between the injection points and the FBG sensors is comparatively larger. Therefore, smaller strain magnitudes are expected and observed.

**Reviewer Point P 1.33** — Page 32, Line 33: Does this actually cast doubt on stress orientation from well data only or is the 30-degree rotation result from another process

such as pre-existing weaknesses affecting the propagation of the induced fractures? I would like to see a discussion on the other arguments for this observed orientation difference between the seismic cloud and the stress orientation from borehole data, with arguments for and against each before coming to a final conclusive statement such as the one currently made.

**Reply**: We revised Section 6.2 and focused the discussion on the difference HF borehole traces and the relation to the frac cycles (i.e. different flow rates). A discussion on stress estimates from wellbore traces using HF or misinterpretation of stress data will not be added. This is out of the scope of this paper and does not help to reduce the manuscript length.

**Section 6.2 Borehole trace and fracture tortuosity:** The comparison of the fracture trace orientation at the borehole wall observed by acoustic televiewer logging with the orientation of the seismic cloud associated with a given fracture allow assessing fracture rotation, also referred to as tortuosity, in the near field of the borehole (Bunger and Lecampion, 2017). An angular difference of about $30°$ is typical for most of our experiments. The inflating packers isolate the injection zone, therefore the HF initiate in the region controlled by near-wellbore stress concentration. The trace of the fracture at the borehole wall will thus be strongly influenced by near-well effects, including borehole deviation compared to the local stress field principal orientation, transverse isotropic mechanical properties and local stress heterogeneity. This brings significant uncertainty on the interpretation of fracture trace at the borehole wall to characterise the in-situ stress state. The minifracs indicate a bigger variability for the trace orientations than for the seismic cloud data (Fig. 14). These experiments were primarily performed in intact rock in two horizontal and one sub-vertical borehole and affected a small volume of intact rock around the borehole. For the HF experiment larger amount of fluid was injected in a moderate fractured rock mass and the flow rate was around 5 l/min during the frac cycle. The variability tend to be smaller for the HF traces using a higher flow rate promoting dominant fluid pathways compared to low flow rate (MF), which

tend to have different fracture geometries in the near-wellbore. The big flow rates and the test volume interacting with the hydraulic fracture will inevitably reach some significant features that affect the geometry of the seismic cloud, which has been confirmed by our experiments (more in the following section). The fracture traces observed at the borehole wall mostly extended below the packers. It is unclear if they initiated at the packer location or grew beyond them later during the successive propagation phase of the experiments.

**Reviewer Point P 1.34** — Page 33, Line 1: How is the trace of a fracture scattered? I encourage the authors to revise the clarity with which they have written sections and statements of their discussion as they are often difficult to follow, and difficult to relate to any conclusion or argument being made.

**Reply**: We agree that fracture scatter is not a well chosen terminology here. What we want to describe here is the variability in fracture trace orientation. We revised the manuscript accordingly as well as rework part of the discussion and conclusions in order to augment clarity and help the reader to relate arguments and conclusion being made. For the same reason, we also reordered the conclusion section.

**For easier reading we change the ordering of the conclusions:**

[revised manuscript text omitted]

**Reviewer Point P 1.35** — Page 33, Line 5-8: I am unsure why the authors state the trace of the induced fractures are complex, they are actually quite clear in form and morphology from the images provided in figures. What is more complex is where these fractures nucleated during HF tests, i.e. at the abutting natural fractures or within the sealed section of well itself? The interaction of the induced fracture with natural fractures outside the packer would seem to infer that the fracture propagated vertically until it met a pre-existing structure and then ceased to propagate vertically. However, that is difficult to prove without directly observing the induced fracture growth. Finally, the comment on stress orientations from induced features being misleading in deviated wells is old news and there are numerous studies (Hickman, McNamara, Davatzes, Barton, Zoback) that have addressed this issue in a number of geothermal well scenarios. For example, McNamara et al., 2015 (Rotokawa geothermal borehole imaging paper) calculated minimal effect on using induced features for stress orientation

in geothermal wells deviated at angles up to ∼22 degrees from vertical. I would like to see the authors acknowledge some of the points raised here in this section of the discussion as they have important implications to their work.

**Reply**: This comment is now addressed and the manuscript is amended accordingly (see the reply to comment P 1.33)

**Reviewer Point P 1.36** — Page 33, Line 11: to some 'extent'.

**Reply**: Fixed.

**Reviewer Point P 1.37** — Page 33, Line 11-13: Have the authors considered that the induced fractures do not grow in a penny-shape fashion due to the influence of the curved free-surface of the borehole wall?

**Reply**: Yes, the authors know that a penny-shaped fracture assumption is a strongly idealized case for various reasons. As pointed out by the reviewer, one of these reasons is the near-well stress perturbation induced by the presence of the borehole itself. Such perturbation is effective primarily over a distance of 2-3 borehole diameter, i.e. about 30 to 40 cm from the borehole in our case (borehole diameter is 146 mm). Our seismic data don't have the resolution to resolve this near-well fracture geometry. In our discussion in Section 6.3 pointed out by the reviewer, "seismic response and fracture geometry", we assess the fracture geometry at a larger scale. At this scale, we interpret the deviation from an idealized radially growing penny-shaped fracture being largely due to the presence of intersecting natural fractures that leading to flow channeling.
We adjusted the introductory sentence of the section to precise the observation scale relevant in this discussion as following:

**Page 34, Line 29-31:** A key element to track fracture geometries beyond the borehole wall traces is microseismic event clouds. The spatial resolution of our seismic data do not permit to map near well effect, but allow assessing the fracture the fracture geometry at a scale >1 m.

**Reviewer Point P 1.38** — Page 34, Line 22-34: It is standard practise to assume that induced features in borehole images over small well lengths are not representative of the regional far field stress orientations but are in fact likely representative of localised effects or heterogeneities of the stress field orientation. This is reflected in the quality ranking system developed for such data by the World Stress Map project. So, while the data presented here by the authors may not agree or align with expected regional far field orientations they are likely still providing accurate information on the local perturbations in the orientation.

**Reply**: We agree with the reviewer and recommend the paper by **?**. The paper presents the extensive rock stress characterization campaign of the In-situ stimulation and circulation (ISC) project, which took place prior to the hydraulic stimulation experiments. The paper indicates the importance to account for transversal isotropic rock models and results in the unperturbed and perturbed stress state. This study is focused on the hydromechanical data from the hydraulic fracturing experiment. We link the different response in rock deformation, fracture fluid pressure, fluid recovery results with the different shear zones (associated with fractures) and local stress states, which is quite stable and can be described by the perturbed stress state.

**Reviewer Point P 1.39** — Page 34, Line 29: Please check the English of this sentence.

**Reply**: We corrected the sentence and changed wording in the following sentence.

**Page 36, Line 14-16:** For experiments executed south or north of S3 the ISIP stabilize around 5 MPa and only deviates slightly. The jacking pressure is similar in magnitude compared to the ISIP for the experiments south of S3 (see Fig. 5). The situation differs for the experiments north of S3 as the magnitude of the jacking pressure is declined.

**Reviewer Point P 1.40** — Page 36, Figure 16: Please include a key for the coloured symbols on the Mohr space diagrams.

**Reply**: We added a color and symbol key to the figure b) and indicated this in the caption.
**The following change is made in the caption of Figure 14:**
... and the best fit plane from seismic (squares) are presented with computed normal stress and shear stress. The color and symbol key in figure b) is as well applicable for figure c).

**Reviewer Point P 1.41** — Page 36, Line 4: Conclusion 4 and conclusion 2 sound contradictory. 2: The growth of the hydraulic fractures is strongly influenced by natural fractures. 4: the spatial distribution of microseismic events associated with the fracture growth seems to be predominantly controlled by the stress state. While I appreciate these are subtly different data being measured I would encourage the authors to comment somewhere on which aspect, if any, is more controlling on fracture growth (stress or pre-existing weaknesses), or discuss somewhere the link between the stress, natural fracturing, and the effects this can have on induced fracture propagation.

**Reply**: The reviewer highlights here an important output from our experiment: the illustration of the competing role of rock mass structuring (natural fractures) and the stress state on hydraulic fracture growth. Our results show that at small scale, details of the fracture growth is complex and largely influenced by the presence of

natural fractures and other heterogeneity. We can highlight this level of details with the analyses of our stain data. At larger scale however, the fracture segment linkage lead to an overall orientation of the active fracture network (newly created hydraulic fractures and reactivated natural fractures) that is largely controlled by the orientation of the stress field. At this scale our micro-seismic events cloud is illuminating the active fracture network. Such effects have already been observed in the past in during hydraulic fracturing experiment with mine back performed at Northparkes Mine in Australia (**?**).

We reorganised and rewrite the conclusion in order to better explain this important observations.
* * *
**Reviewer 2**

**Reviewer Point P 2.1** — The manuscript is well written, but very long. Is there any way to shorten it?

**Reply**: We agree, that the manuscript is long. This is due to the unusual data set, whereby only part of the results could be presented and discussed. Only a small part of the data recorded by the HF experiments can be presented in this manuscript. In addition, it is important to present key data and to describe essential observations. Accordingly, shortening the manuscript is also associated with omitting observations. We have already taken certain shortenings and changes (see the reply to comment P 1.1, 1.32 and 1.33).

**Reviewer Point P 2.2** — The colors in Fig. 8 are not adequate! Many dots are hardly

visible.

**Reply**: We used a color saturation scheme, that is why some of the circles are hardly visible. The more intense color is used for events located farther away from the injection point and linearly decreasing the color saturation towards the injection points. We refer to the supplementary information for a better overview, where each experiment is presented in map and side view for the seismic event location.

**Reviewer Point P 2.3** — One may criticize the discussion section by nothing that there is very little quantitative exploitation of the data. For instance, it would be nice to relate the transmissivity increase to the created fractures in a quantitative way.

**Reply**: The quantitative data are presented in the sections "Overview of the HF experiment" and "Comparison between stress characterization and HF experiments". Because of the length of the manuscript, we decided not to include more discussion topics and summary figures in the discussion. The aim was to discuss the presented results and made the comparison between the minifracs and HF experiments. We rearranged certain part of the discussion for better readability (see the reply to comment P 1.32, 1.33 and 1.39).

[revised manuscript text omitted]